

# On the spatio-temporal representativeness of observations

Nick Schutgens[1], Svetlana Tsyro[2], Edward Gryspeerdt[3], Daisuke Goto[4], Natalie Weigum[1],
Michael Schulz[2], and Philip Stier[1]

[1]Department of Physics, University of Oxford, Parks road, OX1 3PU, England
[2]Norwegian Meteorological Institute, P.O.Box 43 Blindern, Oslo, NO-0312, Norway
[3]Institute for Meteorology, Universität Leipzig, Stephanstr. 3, 04103 Leipzig, Germany (now at Space and Atmospheric Physics Group, Imperial College London, London, SW7 2AJ, United Kingdom)
[4]National Institute for Environmental Studies, 16-2 Onogawa, Tsukuba, 305-8506, Japan

*Correspondence to:* Nick Schutgens (schutgens@physics.ox.ac.uk)

**Abstract.** The discontinuous spatio-temporal sampling of observations has an impact when using them to construct climatologies or evaluate models. Here we provide estimates of this so-called representation error for a range of time and length-scales (semi-annually down to sub-daily, 300 to 50 km) and show that even after substantial averaging of data significant representation errors may remain, larger than typical measurement errors. Our study considers a variety of observations: ground-site remote sensing or in-situ ($PM_{2.5}$, black carbon mass or number concentrations), satellite remote sensing with imagers or LIDARs (extinction). We show that observational coverage (a measure of how dense the spatio-temporal sampling of the observations is) is not an effective metric to limit representation errors. Different strategies to construct monthly satellite L3 data are assessed and temporal averaging of spatially aggregated observations (super-observations) is found to be the best, although it still allows for significant representation errors. Temporal collocation of data (only possible in the context of evaluating model data with observations) can be very effective at reducing representation errors even when spatial sampling issues remain (e.g. when using ground-sites). We also show that ground-based and wide-swath imager satellite remote sensing data give rise to similar representation errors although their observational sampling is different. Finally, emission sources and orography can lead to representation errors that are very hard to reduce even with substantial temporal averaging.

## 1 Introduction

The intermittent temporal sampling and limited field-of-view of observations reduce their representativeness for the actual weather or climate system they are intended to explore. Yet relatively little work has been done on estimating these sampling impacts and how to mitigate them. At the root of this issue lies the spatio-temporal variability of the natural system, but the large variety in sampling strategies of observing systems adds significantly to the complexity of the problem. A *representation error* can be used to describe the ability of measurements to represent a larger area over an arbitrary (but specified) length of time. If the observations are used to evaluate models, these *represented* areas would coincide with the model's gridboxes.

Hakuba et al. (2014b, a) studied the spatial representativeness of ground-sites for solar surface radiation measurements and Bulgin et al. (2016) parametrised the sampling uncertainty in gridded SST (sea surface temperature) measurements (cloud masked) from satellite. Climate statistics were shown to differ between point data and gridded data in theoretical studies by



Cavanaugh and Shen (2015) and Director and Bornn (2015). Recently, Diedrich et al. (2016) studied the impact of cloud-masking in water vapour measurements from satellite and found a 25% lower monthly global mean water vapour path.

In this paper, we will focus on aerosol but our results can be expected to have wider implications. Since the landmark study by Anderson et al. (2003) we know aerosol varies over hours and tens of km, see also Kovacs (2006); Santese et al. (2007); Shinozuka and Redemann (2011); Schutgens et al. (2013); Weigum et al. (2016). Aerosol studies are likely to show a very clear impact from spatio-temporal sampling.

Kaufman et al. (2000); Smirnov (2002) and Remer et al. (2006) attempted to assess the impact of diurnal cycles on the representativeness of satellite observations for daily averages. Similarly, Sayer et al. (2010) and Geogdzhayev et al. (2014) estimated the impact of satellite sampling on monthly and yearly regional averages. These studies showed that significant differences might result from temporal sampling alone. Levy et al. (2009) studied different algorithms to create monthly MODIS (MODerate resolution Imaging Spectro-radiometer) gridded data (so-called L3) and showed large differences might result. A major issue for Levy et al. (2009) was the absence of an objective truth.

The term representation error (or representativity or representativeness error) is often used in data assimilation where a growing body of research exists, e.g. Desroziers et al. (2005); Waller et al. (2014); Hodyss and Nichols (2015); van Leeuwen (2015); Waller et al. (2016). In data assimilation the representation error concerns very short time scales: observations are compared against model data at specific times. In this paper we are also interested in representation errors after averaging over months or even years. Conceptually representation errors in data assimilation have evolved to include model errors due to poorly represented sub-grid processes. In this paper, we are only concerned with the spatio-temporal representativeness of observations.

In two recent studies, we explored temporal and spatial sampling issues using aerosol models as a truth. In Schutgens et al. (2016a) (henceforth S16a) spatial sampling issues for model grid-boxes of a few 100's of kms were explored on time-scales of hours to a month using high-resolution model data. In Schutgens et al. (2016b) (henceforth S16b) temporal sampling issues were explored on time-scales of days to a year using global model data and actual temporal sampling from remote sensing datasets. Major conclusions were: 1) representation errors can be large and are often larger than measurement errors and may be similar to model errors; 2) representation errors can be reduced through spatio-temporal averaging *under certain conditions* and not as much as is commonly implicitly assumed; 3) representation errors vary greatly, depending on the observing system and the observable. Both intensive (e.g. single scattering albedo) and extensive (e.g. aerosol optical depth) observables suffer from representativeness issues.

In S16a, we assumed that observations were made continuously in time. Clearly this is unrealistic for many and in particular remote sensing observations (but it is often a fair assumption for ground-site in-situ measurements). In the current paper, we will study the combined impact of spatio-temporal sampling on representation errors for a wide variety of observing systems (ground-site in-situ, ground-site passive remote sensing, satellite passive and active remote sensing) on a range of time-scales from hourly to semi-annually. Firstly, this will yield more realistic representation error estimates than were previously (S16a a& S16b) possible; secondly, it elucidates the interplay of spatial and temporal sampling in creating representation errors; thirdly,





it explores various strategies in reducing these representation errors. In particular, we will show how temporal collocation of model data with observations can reduce representation errors in model evaluation.

Section 2 describes the high-resolution model data and how they were used to create simulated observations. Section 3 explains how representation errors are calculated from these data. Results for semi-annual averages (Sect. 4), monthly averages (Sect. 5), daily averages (Sect. 6) and sub-daily data (Sect. 7) follow. The impact of precipitation on sampling issues is discussed in Sect. 8. An overview of results per considered observing system is given in Sect. 9 and the paper concludes with a summary (Sect. 10)

Note that Sect. 3.2 contains some general guidelines to interpreting many of the figures and statistics that appear in this paper.

## 2 The regional models

The same simulations as in S16a are used in the current study and for details we refer to that paper. Briefly, the models WRF-Chem (Grell et al., 2005; Fast et al., 2006), EMEP/MSC-W (Simpson et al., 2012) and NICAM-SPRINTARS (see Goto et al. (2015) and references therein) were used to simulate common observables (aerosol optical thickness, extinction, $PM_{2.5}$, black carbon mass concentration, number densities and cloud condensation nuclei) on a 10 km grid with hourly resolution. All models nudged windspeeds to reanalysis meteorology and used emissions with diurnal profiles where relevant. Fig. 1 shows the simulation regions, and Table 1 summarises the most important information on these simulations.

As precipitation is potentially a major cause of spatio-temporal variability in aerosol, we evaluated the models against GPCP (Global Precipitation Climatology Project, Adler et al. (2003); Huffman et al. (2009)) 1-degree daily combination v1.2 data (Huffman et al. (2001), see also http://precip.gsfc.nasa.gov/gpcp_daily_comb.html). Histograms of daily precipitation in the models compare quite well to these observations, see Fig. 2). At higher daily precipitation, there is quite a bit of statistical noise due to the low number of cases, as can be seen by comparing the observation over W-Europe and Europe. The most notable differences from the observations are found for Congo, where the model tends to overestimate precipitation, and Ocean & Japan, where the models tend to underestimate low precipitation cases.

### 2.1 Observable parameters

The simulated fields examined in this paper are, for obvious reasons, all observables, see Table 2. All of the models provided AOT (Aerosol Optical Thickness), AE (Ångström Exponent), SSA (Single Scattering Albedo), extinction and (dry) $PM_{2.5}$, although WRF-Chem calculates AOT and extinction for 600 nm and EMEP and NICAM-SPRINTARS for 550 nm. WRF-Chem MADE provided CCN (Cloud Condensation Nuclei) at varying degrees of super-saturation $S$. Converting WRF-Chem output into observables of black carbon concentration or number densities required some further assumptions that are detailed in S16a.





The spatio-temporal sampling of real observations is determined by their operational parameters and by adverse conditions. For simplicity's sake, we created a number of idealised scenarios for different observing systems. Additional model information like local times, cloud fraction and precipitation were used to create spatio-temporal samplings for the observations.

Ground-site in-situ measurements are assumed to occur at all times, irrespective of conditions but constrained by operational
parameters, e.g. IMPROVE (Interagency Monitoring of PROtected Visual Environments) measures only a full day every three days. Note that this is a best case scenario and most ground-sites will suffer down-time due to maintenance or malfunction. In particular we assume that these measurements will occur irrespective of precipitation since this usually does not prevent measurements. Obviously, in-situ ground-sites only observe a small part (here 10 by 10 km) of the atmosphere near the surface.

Ground site remote sensing observations of AOT will occur during the day-light portion of each day (here 10 hours straddling
local noon), provided there are no clouds. These ground-sites will observe only a small portion (10 by 10 km) of an atmospheric column. Again, down-time due to maintenance or malfunction is not considered.

Passive satellites measurements (imager data) on polar orbiting satellites are assumed to occur once during a day at local noon, provided there are no clouds. Imagers will have swaths wide enough to allow aggregation of individual measurements over the represented area. Due to its orbital parameters and swath width, these satellites will have repeat cycles of 1, 2, 4 or
8 days. Imagers on geostationary satellites allow measurements during the day-light portion of each day (10 hours straddling local noon).

Satellite LIDAR (LIght Detection And Ranging) measurements observe a narrow north-south transect (see also S16a) within the represented area once a day at local noon with a repeat cycle of 12 days. CALIOP (Cloud-Aerosol LIDAR with Orthogonal Polarization) has a repeat cycle of 16 days but allowing the LIDAR swath to revisit different parts of the same 210 by 210 km$^2$
area brings the typical cycle down to about 12 days. As we do not consider measurement errors, it matters little if the LIDAR measurement is made during the day or night. Down-time due to malfunction is not considered.

## 3   Simulating observational and global model data

This section briefly describes the main methodology used in this paper. The high-resolution regional model data $v$ can be thought of as 3-dimensional data cube $v_{xyt}$ (either a column or layer property) where $x = 1 \ldots n_x$ and $y = 1 \ldots n_y$ are indices
to the horizontal coordinates, and $t = 1 \ldots n_t$ is an index to the time coordinate. As the model data has been transformed to a regular grid, equations can conveniently be written down with references to indices only. Using this data cube $v_{xyz}$, we will generate both a truth (an average over a wider area that is to be represented) and a sampled but otherwise noiseless (i.e. without measurement error) observation.

At a single time, the truth for a represented area can be written as

$$30 \quad T_{xyt} = \frac{1}{(2L_x+1)(2L_y+1)} \sum_{i=-L_x}^{+L_x} \sum_{j=-L_y}^{+L_y} v_{x+i;y+j;t}, \tag{1}$$



where $L_x$ and $L_y$ define the half-lengths of the represented area. A time average of this is given by

$$\bar{T}_{xyt} = \frac{1}{2L_t+1} \sum_{k=-L_t}^{+L_t} T_{x;y;t+k}, \tag{2}$$

where $(2L_t + 1)$ defines the averaging period. Note that a capital variable name denotes a spatial average and an overbar a temporal average.

5   In a very similar way, a spatio-temporal average of the observations may be written as

$$\bar{O}_{xyt} = \left( \sum_{k=-l_t}^{+l_t} \sum_{i=-l_x}^{+l_x} \sum_{j=-l_y}^{+l_y} f_{x+i;y+j;t+k} \right)^{-1} \times$$

$$\sum_{k=-l_t}^{+l_t} \sum_{i=-l_x}^{+l_x} \sum_{j=-l_y}^{+l_y} f_{x+i;y+j;t+k}\, v_{x+i;y+j;t+k}, \tag{3}$$

where $l_x, l_y$ and $l_t$ serve a similar purpose as $L_x, L_y$ and $L_t$. The observational sampling $f_{xyt}$ is defined as:

$$f_{xyt} = \begin{cases} 0 & \text{if } \textit{no} \text{ observation present at } x, y, t \\ 1 & \text{if observation present at } x, y, t. \end{cases} \tag{4}$$

Note that this is a very general formulation that can be used to simulate both individual ground-sites and satellite measurements.

The relative spatio-temporal representation error in an observation for arbitary time and length-scales is now given by

$$\bar{\epsilon}_{xyt} = \left( \bar{O}_{xyt} - \bar{T}_{xyt} \right) / \bar{T}_{xyt}. \tag{5}$$

When observations are used to evaluate models, it is possible to temporally collocate model data with observations. We simulate
15   this by constructing $\bar{T}_{xyt}$ from a sub-sampled number of $T_{xyt}$ and the resulting error will be called "representation error with collocation".

Note that it is possible to aggregate observations spatially before temporally averaging them:

$$O_{xyt} = \left( \sum_{i=-l_x}^{+l_x} \sum_{j=-l_y}^{+l_y} f_{x+i;y+j;t} \right)^{-1} \times$$

$$\sum_{i=-l_x}^{+l_x} \sum_{j=-l_y}^{+l_y} f_{x+i;y+j;t}\, v_{x+i;y+j;t}. \tag{6}$$

This is sometimes called super-obbing and the resulting data super-observations. Temporal averages can then be generated from

$$\bar{O}_{xyt} = \left( \sum_{k=-l_t}^{+l_t} G_{xy;t+k} \right)^{-1} \sum_{k=-l_t}^{+l_t} G_{xy;t+k} O_{xy;t+k}, \tag{7}$$

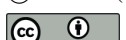



where $G_{xyt}$ defines a sampling, much like $f_{xyt}$. While $f_{xyt}$ will depend on retrieval conditions (e.g. cloudy or not), $G_{xyt}$ is an arbitrary choice (wether to accept a given $O_{xyt}$ as a valid super-observation). The resulting $\bar{O}_{xyt}$ in Eq. 7 is similar to many L3 products for satellite imagers.

Actually, the two expressions for $\bar{O}_{xyt}$ may be related to alternative averages that were proposed by Levy et al. (2009) for satellite L3 products. Their "Pixel Weighting" procedure corresponds to Eq. 3, while procedures "Equal Day Weighting" and "Threshold Equal Day Weighing" correspond to Eq. 7. The difference between the latter two is in the construction of $G_{xyt}$ (requiring a minimum number of pixels for a valid super-observation or not).

To conclude, we introduce three metrics of the abundance of measurements that go into $\bar{O}$ as this will affect how well it compares to the truth. The spatial *coverage* of a single super-observation is

$$c_{xyt}^{\text{spat}} = \frac{1}{(2l_x+1) \times (2l_y+1)} \times$$

$$\sum_{i=-l_x}^{+l_x} \sum_{j=-l_y}^{+l_y} f_{x+i;y+j;t}. \quad (8)$$

The temporal *coverage* of a time-averaged super-observation is defined differently because many observations are made not continuously but nevertheless regularly in time (e.g. satellite overpass times):

$$c_{xyt}^{\text{temp}} = \sum_{k=-l_t}^{+l_t} G_{xy,t+k} / \sum_{k=-l_t}^{+l_t} G_{xy,t+k}^*, \quad (9)$$

where $G^*$ is a sampling entirely defined by the observational cycle of the observing system. This includes orbital and day-light constraints but not cloudiness. Note that in real life, these coverages are known and can be used to select observations; e.g. only aggregated satellite data with a required minimum spatial coverage will be used to compare against model results or only ground-sites with a required minimum temporal coverage will be used to construct monthly averages. (Henceforth we will refer to required coverage and drop the word 'minimum').

Each data cube $v_{xyt}$ will allow us to generate $n_T$ cases of the truth $\bar{T}_{xyt}$. The number $n_O$ of possible $\bar{O}_{xyt}$ cases will be less, depending on both $f_{xyt}$ and $G_{xyt}$. This leads to the definition of a case coverage $n_O/n_T$. Ideally the case coverage is 100% which is possible even if $f_{xyt}$ and $G_{xyt}$ are not always 1 and indicates there are sufficient observations to construct valid $\bar{O}_{xyt}$ anywhere and anytime.

As explained in S16a, the first two days of the high-resolution simulations and the outer part of the spatial domain where excluded from analysis to prevent boundary effects to impact our results.

## 3.1 Some terminology

Representation error will refer to the representativeness of an observation (possibly aggregated over an area and averaged over a time period) in describing the natural system. If observations are used to evaluate temporally collocated model data, we will refer to a representation error with collocation. We will consider two collocation methodologies: to the hour or to the day. In



the first case, *hourly* model data is temporally collocated to the *hour* of the observation. In the second case, *daily* model data is collocated to the *day* of the observation (and the observation itself is a daily average).

### 3.2 Common characteristics of the figures in this paper

This paper contains many figures of representation error distributions. Instead of repeating the same information in each
5 caption, some aspects of those figures are explained here. We use the so-called parametric 7-number summary of the 2, 9, 25, 75, 91 and 98% quantiles $q$ of the errors because, for a normal distribution, these quantiles will be equally spaced. Any skewness or extended wings in a distribution will be readily visible. In addition to quantiles, we will provide RMSD (root mean square differences) and RMSE (root mean square errors, essentially RMSD after removing any bias).

### 3.3 Figures with grey shading

In Fig. 5 different shades of grey are used to denote these interquantile ranges: light grey for $q_{98} - q_2$, medium grey for the $q_{91} - q_9$ and dark grey for $q_{75} - q_{25}$. The solid blue line represents the median error.

### 3.4 Figures with box-whiskers

In Fig. 7, box-whisker plots are shown of the error distributions for each of the regions. Different widths of the bars are used to denote different inter-quantile ranges: narrow for $q_{98} - q_2$, medium for $q_{91} - q_9$ and wide for $q_{75} - q_{25}$. The black rectangle
represents the median error and the black circle the mean error. On top of each bar, the RMSD is shown. The colours of the bars refer to different experiments and are explained in the caption of each separate figure. If a required spatial or temporal coverage was used, this will be shown in the lower left and right corners of the figure. Case coverages per region are shown just above the region names.

In Fig. 9 error distributions for two different experiments are shown side-by-side (much like a violin plot), for each region
and usually as a function of an independent parameter (e.g. represented area size in this example). The values above each box-whisker is the ratio of the right error distribution's RMSD to the left one's.

### 3.5 Figures with line graphs

A very different figure is Fig. 8 where error statistics are summarised as a function of required spatial or temporal coverage. The coloured lines represent RMSE (solid) and bias (dashed) using the *left*-hand axis. The colours are identical to the ones
used in the box-whisker plots to help identify different experiments. The black lines use the *right*-hand axis and denote the case coverage (solid), and achieved spatial (dashed) and temporal (dotted) coverage. The latter have of course been averaged over all relevant cases.





## 4 Representativeness of semi-annual data

The EMEP simulation, Table 1, allows us to explore sampling issues in semi-annual data, assuming ground-sites representing an area of $210 \times 210$ km$^2$. Figure 3 shows relative representation errors in AOT and surface black carbon mass concentrations. The surface black carbon measurements are continuous through the 6 months while the AOT measurements are only made during day-time and cloud-free conditions, see Sect. 2.1.

Representation errors in surface black carbon measurements are clearly related to emissions sources (notice major cities like Paris and Madrid) and orography (notice the Alps, the Apennines and the Carpathian mountains). On the other hand, representation errors in AOT are dominated by temporal sampling and show a clear region-wide bias as clear-sky day-light AOT tends to be lower than average AOT. In both cases, representation errors can be several 10's of percent. If the AOT measurements are used for model evaluation, temporal collocation of model data to the observations (as advocated in S16b) is possible and the errors are reduced significantly. In particular, the region-wide bias is much reduced and the remaining error pattern is more similar to that for black carbon, see Fig. 4.

Table 3 shows representation errors for several ACTRIS (Aerosol, Clouds & Trace gases Research Infra-Structure) sites within the Europe domain, not just for long-term averages but daily RMSD as well. Representation errors driven by spatial sampling often benefit from temporal averaging unlike errors due to temporal sampling. Collocation removes the difference in temporal sampling and allows remaining representation errors to be reduced through temporal averaging. Note that sources and orography can create conditions where temporal averaging is not very beneficial.

The impact of averaging period on *spatial* representation (AOT is now assumed to be measured continuously) can be seen in Fig. 5. This suggests that averaging over less than 10 hours or more than 1000 hours (6 weeks) has little impact on spatial representation errors.

Note that in S16a we showed that the EMEP simulation yielded smaller spatial representation errors than the WRF-Chem simulation (although they agreed in magnitude and spatial patterns).

## 5 Representativeness of monthly data

The following analysis was made for a represented area of $L_x = L_y = 210$ km, with exceptions noted. All data were averaged over a month.

### 5.1 Remote sensing ground-site

We start with the case of a remote sensing ground-site, see Sect. 2.1. Figure 6 shows representation errors for different regions as box-whisker plots. The figure shows that temporal sampling significantly increases representation errors. Over Ocean and Japan, that even lead to region-wide biases. Temporal sampling is dominated by cloudiness, and cloudy AOT (included in the area data) is larger than clear-sky AOT for these regions.





When evaluating models, Fig. 7 shows that temporal collocation of area data with the observations can substantially reduce representation errors. Here we limited ourselves to locations with at least 25% temporal coverage. Note that temporal coverage is a 100% if each day during the month yields 10 hours of observations. Obviously, representation errors after collocation can never be smaller than purely spatial representation errors. Interestingly, collocation to the day is much less beneficial than collocation to the hour, even after averaging over a month.

Figure 8 shows various error estimates as a function of required temporal coverage for two regions that are typical. As a rule, with increasing temporal coverage the case coverage will go down. This means that the number of ground-sites supplying sufficient observations goes down. Representation errors may go down (Japan) but it is also possible they remain constant (Oklahoma). For all regions, collocation to the hour allows smaller representation errors at lower temporal coverage and higher case coverage than no collocation.

Representation errors are remarkably insensitive to the size of the represented area, unless area data can be temporally collocated, see Fig. 9. This is unsurprising as we earlier pointed out that temporal sampling dominates the representation error.

## 5.2 Passive remote sensing measurements from polar orbiting measurements

Next we turn to polar-orbiting satellites measurements with repeat cycles of 1 or 8 days, see Sect. 2.1. For now, we will assume that individual pixel measurements are averaged together (i.e. no super-obbing), see Eq. 3. Fig. 10 shows representation errors for different regions as box-whisker plots. Due to the aggregation of measurements, purely spatial representation errors are zero. But the spatio-temporal errors are substantial. Depending on the repeat cycle, either cloudiness or the observational cycle is more important to these errors, although it is cloudiness that leads to region-wide biases in the errors (see Ocean & Japan). Note also the very similar spatio-temporal representation errors, despite very different spatio-temporal sampling, for a ground-site, Fig. 6, or a satellite with a repeat cycle of 1 day.

The strong impact of cloudiness on temporal sampling and hence representation errors, shown both here and in the previous sub-section, suggests that area data calculated for clear skies only would yield smaller representation errors. This indeed reduces the region- wide biases over Ocean and Japan see for a 1 day repeat cycle, but the representation RMSE are much the same. We will continue to calculate area data as a total sky average.

Figure 11 shows the impact of temporal collocation. Again, collocating area data to the hour yields smaller representation errors than collocating to the day. For longer repeat cycles monthly representation errors after collocating will be larger because there is less data to average out spatial representation errors. Spatial and temporal coverage requirements were set at 25%, implying that at least 25% of the represented area was observed during 25% of the overpasses.

Alternative methods exist to construct monthly observations, for example by temporally averaging super-observations, see Eq. 7. This has a small but beneficial impact on representation errors. Figure 12 shows representation errors when using super-observations, either straight as in Eq. 7 or log-transformed before temporal averaging. Neither method is capable of achieving the small representation errors due to collocation.

Adjusting required temporal coverage has a similar impact as for ground-sites, see Figure 13. Case coverage (percentage of the region observed by the satellite) goes down as temporal coverage increases. But there is no unequivocal impact on





representation errors: they may remain similar (e.g. Oklahoma) or decrease (e.g. Japan). On the other hand, increasing required spatial coverage has a detrimental effect on representation errors. The reason is that increasing spatial coverage is accompanied by reduced temporal coverage which makes the observations less representative for the full month. The obvious exception is representation errors with collocation (to the hour) that decrease with increasing spatial coverage. We conclude that generally

coverage is not a good measure for representation errors but spatial coverage provides a good control on representation errors with collocation to the hour.

Currently satellite super-observation products (L3) for AOT are usually produced at $1^o \times 1^o$ ($110 \times 110$ km$^2$ at the equator). Using such a product to represent the natural system at different spatial scales yields similar representation errors (as temporal sampling issues dominate), see Fig. 14. But when using it to evaluate collocated model data, representation errors can be

expected to be smallest for $1^o \times 1^o$ model grid-boxes. Note that larger grid-boxes may be filled in multiple super-observations, and so reduce representation errors with collocation.

Finally, we return to the work by (Levy et al., 2009) as several of their strategies for calculating monthly L3 data are easily evaluated in the context of our work (Sect. 3). The aforementioned Fig. 12 shows that "Pixel Weighting" (brown) generally allows larger representation errors than "Equal Day Weighting" (dark blue). "Threshold Equal Day Weighting" is studied in

Fig. 13 (dark blue line as function of spatial coverage) and also shown to allow larger errors than "Equal Day Weighting" (which is identical to "Threshold Equal Day Weighting" with $c^{\mathrm{spat}} > 0$). Thus we conclude that "Equal Day Weighting" is, from a spatio-temporal sampling perspective) the best choice. This will nevertheless allow monthly representation RMSD of 10 to 40%.

### 5.3 Passive remote sensing measurements from geostationary satellites

Geostationary satellites with passive remote sensing instrumentation allow for spatial aggregation of observations and multiple measurements per day. Consequently sampling issues are entirely dominated by cloudiness. Figure 15 shows that even for an imager in geostationary orbit, monthly representation errors are quite substantial. Actually, they are not that different from an imager on a polar orbiting satellite (Fig. 11) with a 1 day repeat cycle or a ground-site (Fig. 7). The reason is of-course that cloudiness is the main reason for representativeness issues. Note that representation errors after collocation are substantially

lower for the geostationary imager than for a ground-site but again similar to those for polar-orbiting imager.

### 5.4 LIDAR measurements from polar orbiting satellites

An idealised polar orbiting LIDAR, see Sect. 2.1, allows for limited aggregation (along its track) but will have a long repeat cycle (here: 12 days). Figure 16 shows the resulting representation errors with and without collocation. These errors are large, even with collocation, and may preclude the use of satellite LIDAR data on monthly and 100 km scales. However, further

averaging of *temporally collocated* data over larger regions (say Europe or the Atlantic dust outflow region) is likely to reduce representation errors as they are often not strongly correlated over distances exceeding the size of the represented area (e.g. see Fig. 3 or Fig. 2 in S16a).



### 5.5 In-situ ground-sites

The IMPROVE network operates on a regular schedule of measuring one day out of three. Figure 17 shows that this has a relatively mild impact on representation errors. Still, errors may increase two-fold and collocation will usually bring representation errors down to the level of purely spatial errors. Due to the observing cycle, it doesn't matter whether this is collocation to the hour or day. Similar results can be shown for black carbon concentration or number density measurements.

## 6 Representativeness of daily remote sensing data

The following analysis was made for a represented area of $L_x = L_y = 210$ km, with exceptions noted. All data were averaged over a day.

### 6.1 Remote sensing data

Figure 18 shows daily representation errors for either ground-sites or imagers on polar-orbiting satellites with a repeat-cycle of 1 day. Spatial representation errors are quite large for ground-sites but they are zero for the satellite. Yet spatio-temporal representation errors (without collocation) are very similar (although a bit smaller for the imager). Collocation to the hour reduces representation errors, but more so for the aggregated satellite observations. Actually, collocation for ground-sites allows for still significant spatial sampling issues in daily data.

Typical impacts of observational coverage are shown in Fig. 19. For the ground-sites more stringent conditions on temporal coverage of the observations are relatively ineffective, irrespective of collocation or not: the spatial sampling issue always remains. In model evaluations, collocation to the hour will allow representation errors in satellite data to be arbitrarily reduced by specifying a spatial coverage requirement. Note however that data availability drops steadily as coverage is increased.

The imager on a geo-stationary satellite again shows similar representation errors to the other observing systems with the exception of W-Europe where an RMSD of 20% was found, a significant improvement over ground-sites (37%) and polar-orbiting satellites (29%).

### 6.2 In-situ ground-sites

In-situ ground-sites that observe continuously during the day will have identical daily representation errors, with or without collocation. Here we find daily representation RMSD for $PM_{2.5}$ to range from 7% (Ocean) to 100% (Congo) with most values between 10 and 30%.; and for surface black carbon concentrations 40–100%.

## 7 Improving representativeness for data at less than daily time-scales

Sofar we have tacitly assumed that monthly or daily averages over the representative area are best represented by monthly or daily observations. But at an hourly scale areas may be better represented by longer time averages of the observations,





using wind advection to observe more than the instruments instantaneous field-of-view. Here we will average the represented area over an hour or a day, and see what are the optimal averaging time-scales for the observations (from ground-sites). Remote sensing observations will be treated as uninterrupted by clouds or nighttime, to allow easier comparison to in-situ measurements.

When considering represented areas at daily time-scales, the optimal period for averaging observations (at which the representation RMSD is minimal) is more than a day, see Fig. 20 and Table 4. However, using 24 hours for averaging observations doesn't result in significant increases in representation error and justifies the analysis in Sect.6.

Figure 21 shows hourly representation errors as a function of averaging period of surface $PM_{2.5}$ observations. It is obvious that hourly observations do not guarantee the smallest representation error. Averaging the observations over several hours

results in substantially better representation. There is quite a bit variety in optimal averaging period but it turns out that 6 hours is a good recommendation, also for other observables, see Table 5. This optimal period is firstly the result of a golden middle way: for both short and long periods large representation errors due to spatial or temporal sampling issues may be expected. In between there is a fairly large range of periods (including 6 hours) for which the representation error is close to minimal.

In a few cases optimal averaging periods can be linked to the time needed for aerosol to drift a distance similar to the extent

of the represented area (so-called transit time), see Fig. 22. But this was possible only for a few observables and seldom for surface measurements (N10 at 2 km is the best example we found). We surmise that turbulent flow and evolving aerosol make the link between transit times and optimal averaging periods rather tenuous.

At smaller representative areas of $110 \times 110 \ \mathrm{km}^2$, an averaging period of 4 hours is recommended.

## 8   Impact of precipitation on representation errors for in-situ measurements

Due to its importance in removing aerosol from the atmosphere, precipitation may be expected to be a leading cause of spatio-temporal variability in aerosol. In this section we explore if it is feasible to control representation errors by selecting observations for dry days only.

Precipitation is measured either locally by directly measuring the rain flux (e.g. rain buckets), or regionally through remote sensing measurements (e.g. scanning rain radar). This suggests two potential predictors for the impact of precipitation on

representation errors: 1) a local precipitation measurement sited near the in-situ aerosol measurement can be used to identify cases of strong precipitation; 2) regional measurements can be used to identify cases where precipitation over the ground-site and the wider represented area differ greatly.

Figure 23 shows a rather typical example of how daily representation errors for in-situ measurements correlate with local precipitation. It is obvious that the impact is not overly large considering the already sizeable representation errors at low

precipitation. Most observables and regions show even less dependence on precipitation. Over the Congo, higher local precipitation actually leads to *smaller* representation errors. The second predictor, the relative difference in precipitation over the wider area and at the ground-site, shows even less conclusive results.



Fig. 24 examines how monthly representation errors change due to the discarding of observations with potentially high representation errors (based on the aforementioned predictors). This has only a marginal impact and quite often that impact is to increase representation errors, albeit slightly. This happens because the temporal averaging over less data leads to larger representation errors, similar to what we saw for remote sensing observations. These results do not depend on the chosen

observable, region or (arbitrarily chosen) threshold for the predictor. Only surface aerosol extinction over Japan showed a small but beneficial impact on representation errors due to filtering out high precipitation events. Note that the area data were collocated to the hour with available observations before monthly averaging, to provide a best case.

Concluding, our analyses suggest that no systematic beneficial impact due to discarding cases of high precipitation or strong spatial gradients in precipitation can be found. This holds also at smaller sizes of the represented area (down to $50 \times 50$ km$^2$).

Studying movies of the evolving aerosol in our simulations offers an explanation: precipitation is seldom limited to the ground-site and the represented area will be affected as well; also, precipitation does necessarily correlate with loss of aerosol as converging air motions near updrafts or the sulfate production in associated cloud fields may actually increase aerosol; finally, the spatio-temporal distribution of emission sources combined with changing wind-fields are strong drivers of spatial variability by themselves.

**9   Lessons learned**

While representation errors can be significant, they behave differently depending on whether spatial or temporal sampling dominates the error. In case of spatial sampling, representation errors can often be reduced through spatio-temporal averaging (see also S16a). In the case of temporal sampling, representation errors are unlikely to be reduced through such averaging (see also S16b). If observations are used for model evaluation, it is possible to temporally collocate the model data with the

observations, further reducing representation errors.

Typical representation RMSD errors and other numerical results quoted below refer to a represented area of $210 \times 210$ km$^2$. For other area sizes, see S16a or this paper. For model evaluation, we used a required spatial and/or temporal coverage of 25% and collocation to the hour.

To have observations optimally represent a larger area, they will need to be averaged over time. While monthly area data is

best represented by monthly observations, hourly area data is better represented by observations averaged over 6 hours.

**9.1   In-situ ground-sites**

If such sites allow for continuous operation the measurements from these sites only suffer representation errors due to spatial sampling. Temporal averaging may reduce such errors but emissions sources and orography may cause a constant component in representation error that can not be eliminated. We found errors up to 40% in 6-months averages of surface black carbon

mass concentrations, Sect. 4. We suggest vetting such observations for location.

*For model evaluation:* Averaging both model data and observations over multiple sites can be used to increase representativity (see also S16a).





### 9.2 Passive remote sensing ground-sites

These observations suffer from both spatial and temporal sampling issues and the latter is usually more important. A representation error driven by temporal sampling is unlikely to be reduced through temporal averaging, see Sect. 4 and also S16b. Further study is required to validate the use of such observations to construct climatologies. The number of observations used in constructing monthly averages *cannot* be used to control representation errors, see Sect 5. Representation errors in AOT are typically 10–40% (monthly) and 20–50% (daily).

*For model evaluation:* Collocating model data to the hour of observations should be a first step to reduce representation errors. This also provides a control on such errors through the number of available observations. The representation error due to spatial sampling may be reduced by temporally averaging the collocated data. Representation errors in AOT are typically 5–15% (monthly) and 10–30% (daily). Collocation to the day of observation is sub-optimal; we found very similar representation errors as when no collocation is used, see Sect. 5. See also in S16b how collocation to the day creates a longitude dependence in representation errors.

### 9.3 Passive remote sensing imagers on satellites

These observations suffer from both spatial and temporal sampling issues but often allow spatial aggregation over the represented area. Temporal sampling will dominate representation errors and prove insensitive to temporal averaging, see Sect. 4 and also S16b. Further study is required to validate the use of such observations to construct climatologies. The number of (super-)observations used in constructing monthly averages *cannot* be used to control representation errors, see Sect 5. For imagers on polar-orbiting satellites, monthly representation errors in AOT are typically 10–40% (repeat cycle: 1 day) and 35–55% (repeat cycle: 8 days). Daily representation errors in AOT are 25–40%.

*For model evaluation*: temporal collocation of model data to the hour of super-observations is the best strategy. The collocation provides a control on representation errors through the number of available observations and in principle the representation error due to spatial sampling can be arbitrarily reduced before temporally averaging the collocated data (although it may entail discarding numerous useful data). Monthly representation errors in AOT are typically 5–15% (repeat cycle: 1 day) and 10–15% (repeat cycle: 8 days). Daily representation errors in AOT are 10–15%. This daily representation error is significantly lower than that for ground-sites due to the spatial aggregation. As in the case of remote sensing ground-site observations, collocation to the day of observation is sub-optimal, see Sect. 5.

### 9.4 Active remote sensing satellites

Due to their narrow swath, LIDAR observations from space will have long repeat-cycles causing significant representation errors. Monthly representation errors in aerosol extinction are 70–160% with significantly skewed error distributions. Note that we only considered a single atmospheric level near the top of the boundary layer in our very limited study.

*For model evaluation*: monthly representation errors after collocation to the hour were still 20–40%, although one region





(Ocean) showed errors of 140%. Further reduction of representation errors should be possible by averaging all data over larger geographic regions.

## 10   Conclusions

Measurements always have a discontinuous spatio-temporal sampling, unlike the natural system they are trying to observe. As a consequence, actual daily, monthly and yearly averages over areas may be very different from those based on the undersampled observations. This limits the information present in observations and their usefulness in describing nature and consequently the evaluation of models. In this paper, we have estimated these representation errors using high-resolution models to generate an objective truth and synthetic observations for a slew of idealised observing systems (in-situ ground-sites, remote sensing ground-sites, passive and active remote sensing satellites). For a wide range of time-scales (hour-daily-monthly to semi-annually) and length-scales (50 - 300 km), representation errors were shown to be significant, ranging from 10-100%.

In particular, we study typical aerosol observables like AOT, $PM_{2.5}$, black carbon mass concentrations and number concentrations for idealised observing systems that capture the essence of real-life observing systems like AERONET (AErosol RObotic NETwork), SKYNET, IMPROVE, EMEP (European Monitoring & Evaluation Programme), MODIS, AATSR (Advanced Along-Track Scanning Radiometer), MISR (Multi-Angle Spectro-Radiometer) and CALIOP. Typical length-scales at which we estimate representation errors (100's of kms) are based on the grid-resolution of the global models often used in our field.

Our study not only allows us to estimate representation errors but also assess various ways in which to reduce them. In particular, we were able to assess the usefulness of different methods to generate gridded satellite L3-data (Levy et al., 2009). Our results suggest that the current practice of unconditional averaging of super-observations into a monthly product is a good procedure but still allows for significant monthly representation errors (10–40% at best). Small improvements are possible if the super-observations are log-transformed before averaging.

When using observations to evaluate models, it is possible to temporally collocate model data with the observations and we showed this to be a very efficient way to reduce representation errors, especially if this is followed up by temporal averaging. However, such collocation should use hourly model data collocated to the hour of the observation. Currently, daily model data is often collocated to the day of the observation and this is sub-optimal (and sometimes no better than no collocation). Also, collocation allows some control on representation errors through the number of observations used.

Some other interesting finds are: 1) to better represent hourly data for a larger area, observations should be averaged over 6 hours ($210 \text{ km}^2$) or 4 hours ($110 \text{ km}^2$); 2) representation errors for either remote-sensing ground sites or imagers on polar-orbiting (1 day repeat cycle) or geostationary satellites are very similar on daily and monthly scales, despite very different sampling; 3) representation errors often depend counter-intuitively on observational coverage (the number of observations used); 4) temporal sampling issues clearly dominate representation errors in remote sensing data on monthly scales and less clearly dominate on daily scales; 5) local precipitation does not appear to be a major cause of representation errors, and vetting





observations based on precipitation measurements does not improve representativity; 6) emission sources and orography can give rise to persistent and significant representation errors.

Since we used simulations to assess representation errors, our results depend on the quality of the numerical models. In (Schutgens et al., 2016b) we showed that two different models estimated very similar representation errors over the same region. A more fundamental issue is that we only have simulations over 6 different regions for a few months. Clearly this may not be representative of the entire globe. We surmise that magnitudes of representation errors may be affected by this but their behaviour (e.g. impact from sampling or collocation) need not be. Those results were very robust across all 6 regions and more-over agree with common sense.

It is possible that the representation errors estimated in this paper are under-estimates. As argued in S16a, 1) model variability tends to increase with increasing resolution, 2) at 10 km resolution, we can not resolve the fine-structure at the scale of in-situ sampling volumes, 3) we use assumed temporal profiles of our emission that do not capture day-to-day or week-to-week variations, and 4) our models offer only a bulk abstraction of aerosol without all the detail nature has to offer. At the same time, the use of regional models may preclude proper simulation of pristine regions.

## 11 Code availability

Copies of the code used in our analysis are readily available from the corresponding author.

*Acknowledgements.* This work was supported by the Natural Environmental Research Council grant nr NE/J024252/1 (GASSP: Global Aerosol Synthesis And Science Project).

E. Gryspeerdt acknowledges funding from the European Research Council under the EU Seventh Framework Programme FP7-306284 ('QUAERERE'). D. Goto was supported by the Global Environment Research Fund S-12 of the Ministry of the Environment (MOE)/Japan, the Grant-in-Aid for Young Scientist B (Grant Number 26740010) of the Ministry of Education, Culture, Sports and Science and Technology (MEXT)/Japan, and KAKENHI/Innovative Areas (Grant Number 24110002) of MEXT/Japan. P. Stier and N.A.J. Schutgens acknowledge funding from the European Research Council under the EU Seventh Framework Programme (FP7/2007-2013) / ERC grant agreement FP7-280025 (ACCLAIM: Aerosol effects on Convective CLouds And clIMate).

Michael Schulz and Svetlana Tsyro acknowledge funding from the Norwegian Research Council under the KLIMAFORSK project 'AeroCom-P3'. Their work was supported by EMEP under UNECE.

The figures in this paper were prepared using David W. Fanning's coyote library for IDL.



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





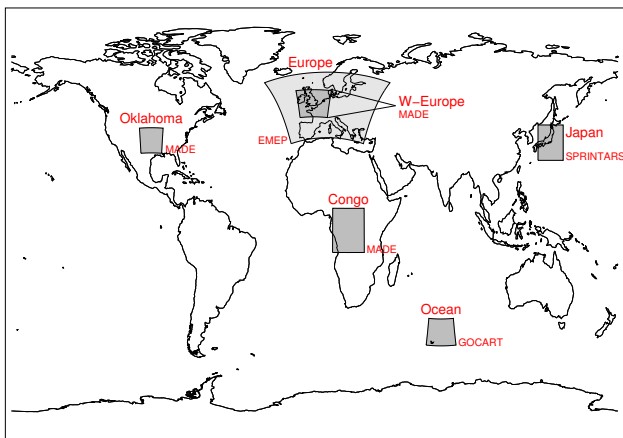

**Figure 1.** Three models were used in this study to simulate a variety of aerosol fields. The regional names used to identify these simulations are given in large font, while the models are denoted in small font. MADE and GOCART refer to the WRF-Chem version used.

van Leeuwen, P. J.: Representation errors and retrievals in linear and nonlinear data assimilation, Quarterly Journal of the Royal Meteorological Society, 141, 1612–1623, doi:10.1002/qj.2464, 2015.

Waller, J. A., Dance, S. L., Lawless, A. S., Nichols, N. K., and Eyre, J. R.: Representativity error for temperature and humidity using the Met Office high-resolution model, Quarterly Journal of the Royal Meteorological Society, 140, 1189–1197, doi:10.1002/qj.2207, 2014.

5   Waller, J. A., Dance, S. L., and Nichols, N. K.: Theoretical insight into diagnosing observation error correlations using observation-minus-background and observation-minus-analysis statistics, Quarterly Journal of the Royal Meteorological Society, 142, 418–431, doi:10.1002/qj.2661, 2016.

Weigum, N., Schutgens, N., and Stier, P.: Effect of aerosol sub-grid variability on aerosol optical depth and cloud condensation nuclei: Implications for global aerosol modelling, Atmospheric Chemistry and Physics Discussions, 0, 113 619–13 639, doi:10.5194/acp-2016-
10  360, http://www.atmos-chem-phys-discuss.net/acp-2016-360/, 2016.





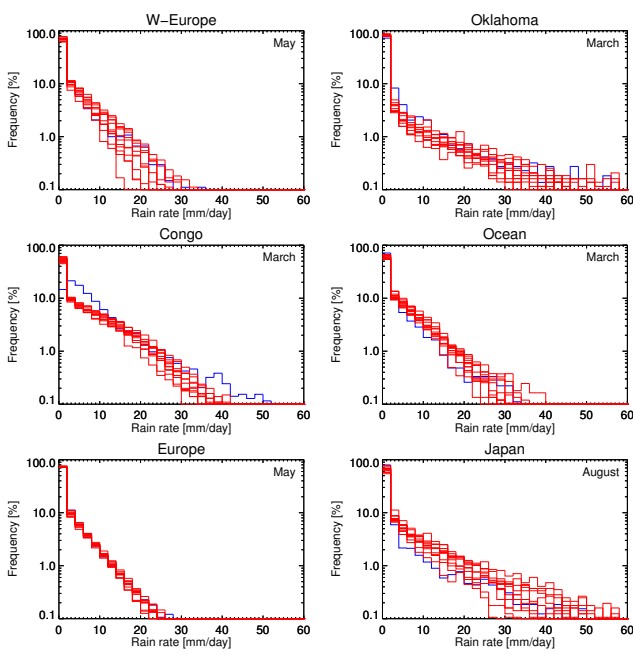

**Figure 2.** Comparison of observed (GPCP) and modelled daily 1-degree precipitation for specific months. The blue line represents the model data (see Table 1), the red line the observations for individual years (2000-2010).





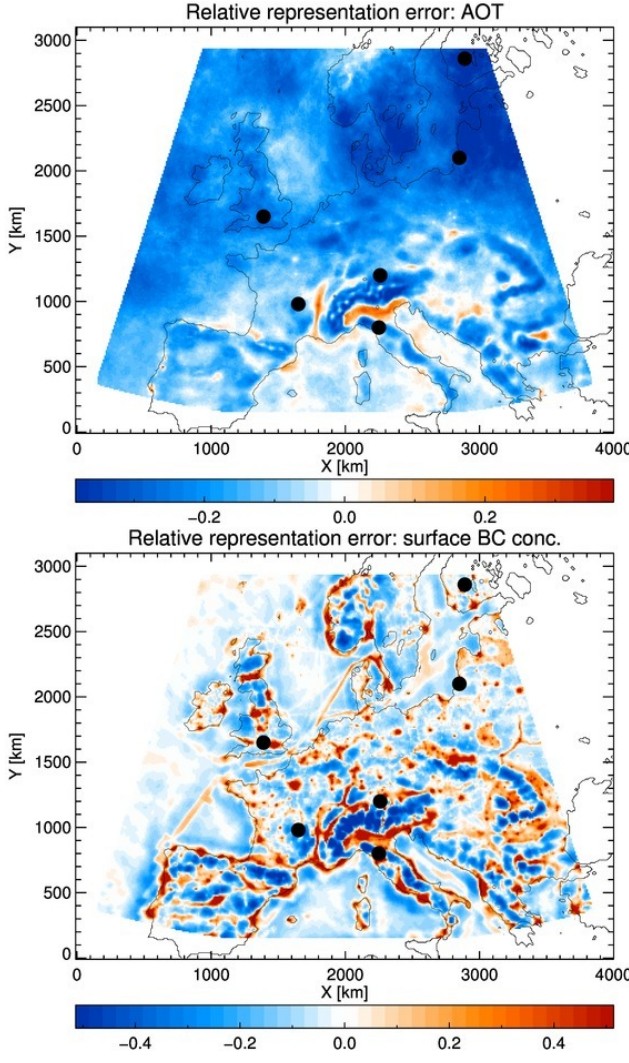

**Figure 3.** Relative representation errors in AOT and surface black carbon concentrations in 6-month averages. The black dots show the locations of major ACTRIS measurement sites. Results for a $10 \times 10$ km$^2$ observation against a $210 \times 210$ km$^2$ area.



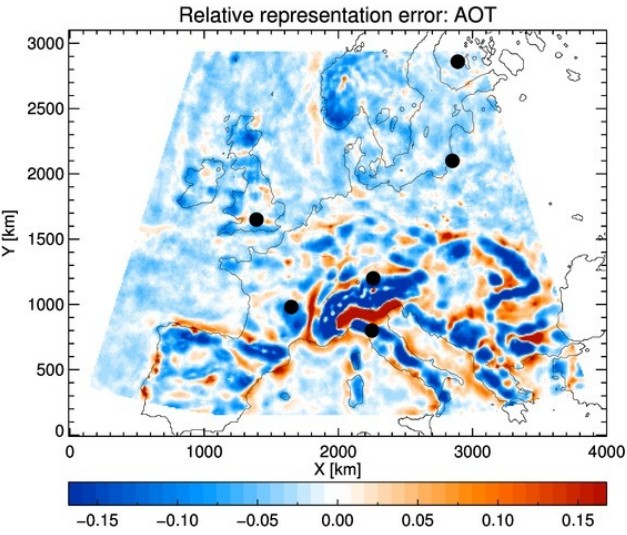

**Figure 4.** Relative representation errors in AOT in 6-month averages. The represented area data were temporally collocated to the hour with the observations. The black dots show the locations of major ACTRIS measurement sites. Results for a $10 \times 10$ km$^2$ observation against a $210 \times 210$ km$^2$ area.

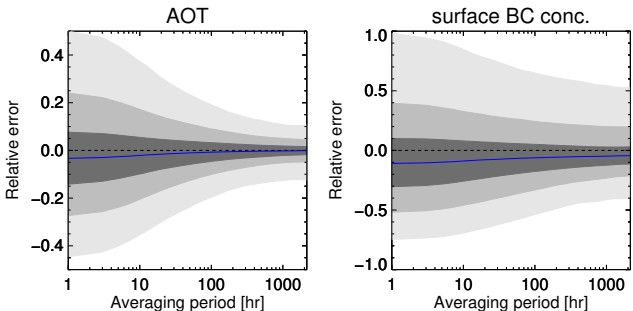

**Figure 5.** Relative *spatial* representation errors in AOT and surface black carbon mass concentrations as a function of averaging period. Both AOT and BC measurements were assumed to be continuous in time. Results for a $10 \times 10$ km$^2$ observation against a $210 \times 210$ km area. Further explanation in Sect. 3.2.





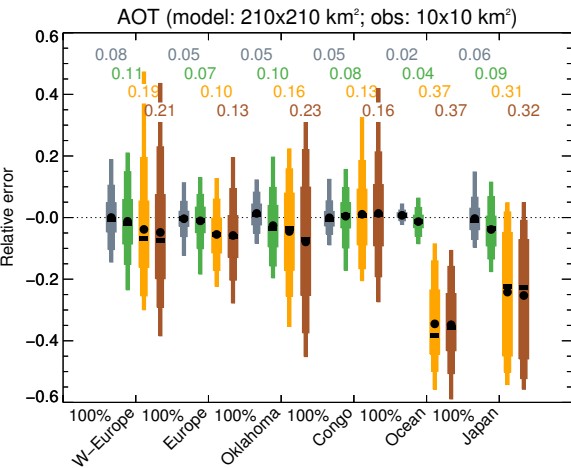

**Figure 6.** Analysis of monthly representation errors for remote sensing ground-sites: purely spatial sampling (grey), spatial sampling and the observational cycle (green), spatial sampling and cloudiness (orange), and finally full spatio-temporal sampling (brown). Further explanation in Sect. 3.2.

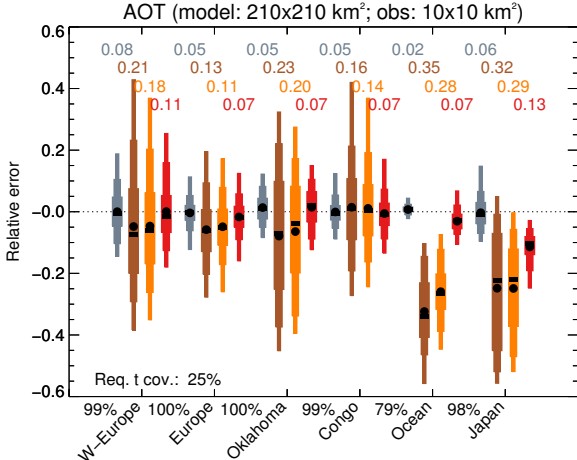

**Figure 7.** Monthly representation errors after collocation for remote sensing ground-sites: purely spatial sampling (grey), no collocation (brown), area data collocated to the day of observations (bright orange), and area data collocated to the hour (red). The grey and brown error estimates are similar to Fig. 6, except for a required temporal coverage of 25%. Further explanation in Sect. 3.2.



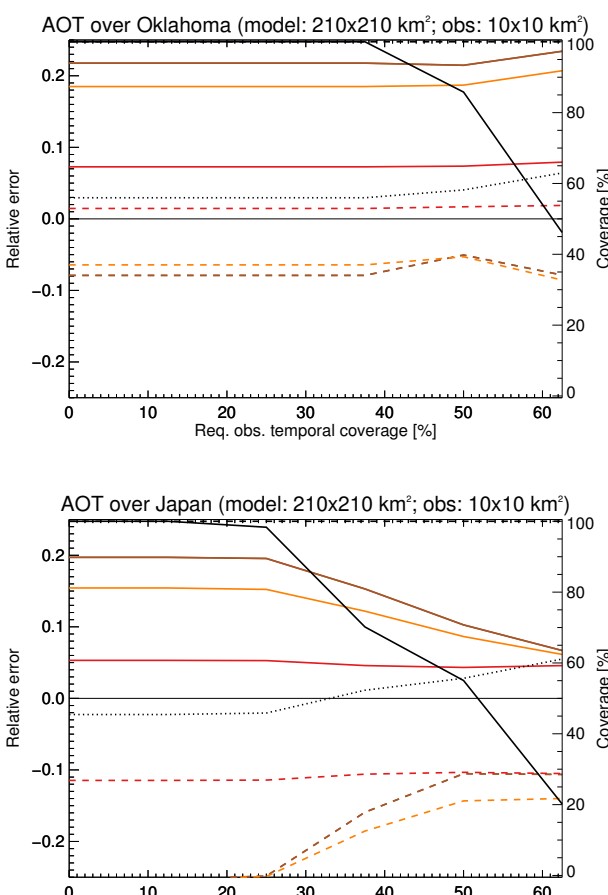

**Figure 8.** Monthly mean (dashed) and RMS (solid) of representation errors for remote sensing ground-sites as a function of required temporal coverage: no collocation (brown), area data collocated to the day of observations (bright orange), and area data collocated to the hour (red). Further explanation in Sect. 3.2.





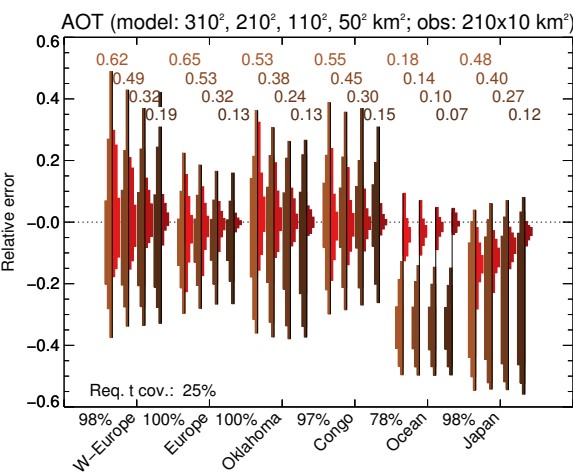

**Figure 9.** Monthly representation errors for remote sensing ground-sites at different area sizes: no collocation (different shades of brown) and model data collocated to the hour (different shades of red). Further explanation in Sect. 3.2.




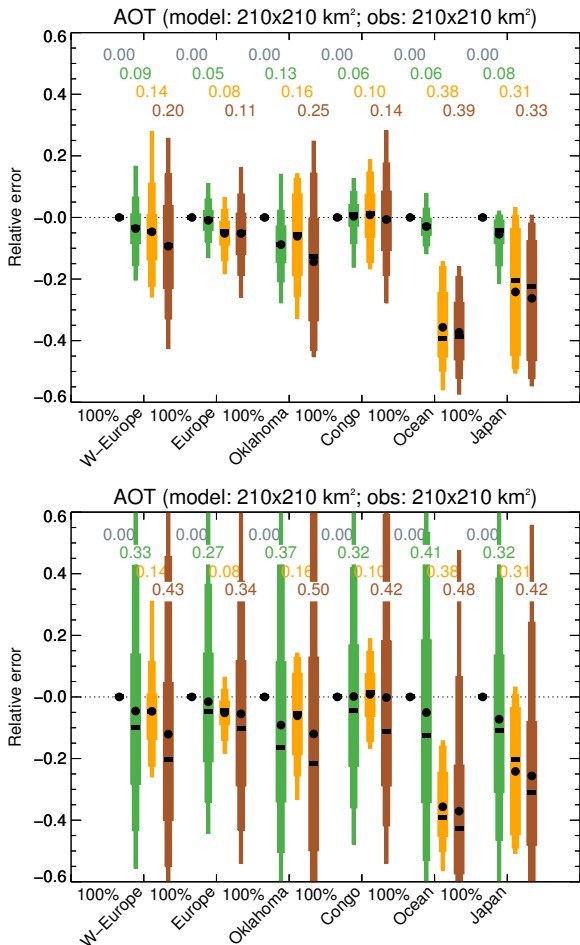

**Figure 10.** Analysis of monthly representation errors for an imager on a polar-orbiting satellite: purely spatial sampling (grey; this error is zero by construction), spatial sampling and the observational cycle (green), spatial sampling and cloudiness (orange), and finally full spatio-temporal sampling (brown). The top panel is for an imager with a repeat cycle of 1 day, the bottom panel for a repeat cycle of 8 days. Further explanation in Sect. 3.2.



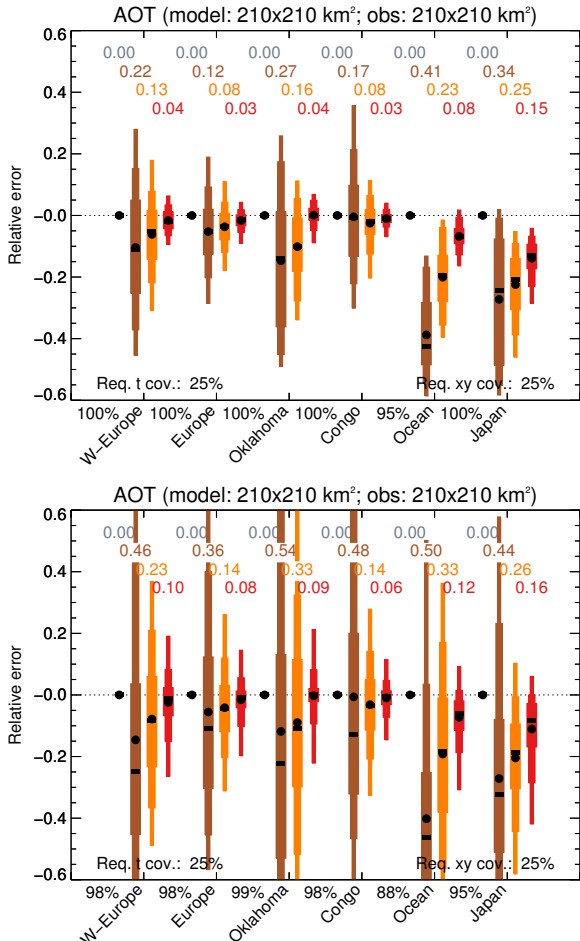

**Figure 11.** Monthly representation errors for an imager on a polar-orbiting satellite after collocation: purely spatial sampling (grey; zero by construction), no collocation (brown), model data collocated to the day of observations (bright orange), and finally model data collocated to the hour (red). The grey and brown error estimates are similar to Fig. 10, except for a required coverage of 25%. The top panel is for an imager with a repeat cycle of 1 day, the bottom panel for a repeat cycle of 8 days. Further explanation in Sect. 3.2.



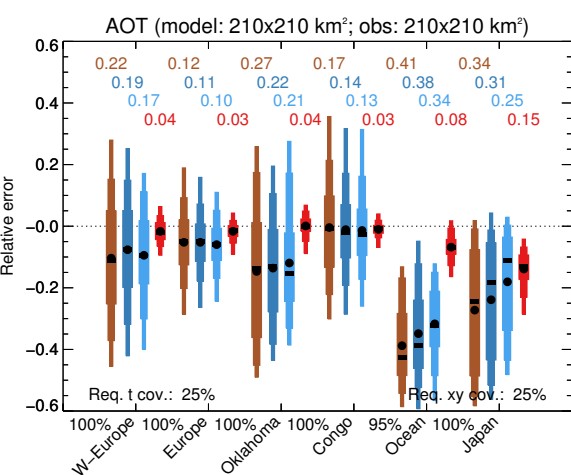

**Figure 12.** Monthly representation errors for an imager on a polar-orbiting satellite due to different data treatments: no collocation (brown), no collocation but using super-observations (dark blue), no collocation but area data and super-observations log-transformed (light blue), and area data collocated to the hour (red). The brown and red error estimates are identical to Fig. 11, top panel. Results for a repeat cycle of 1 day. Further explanation in Sect. 3.2.



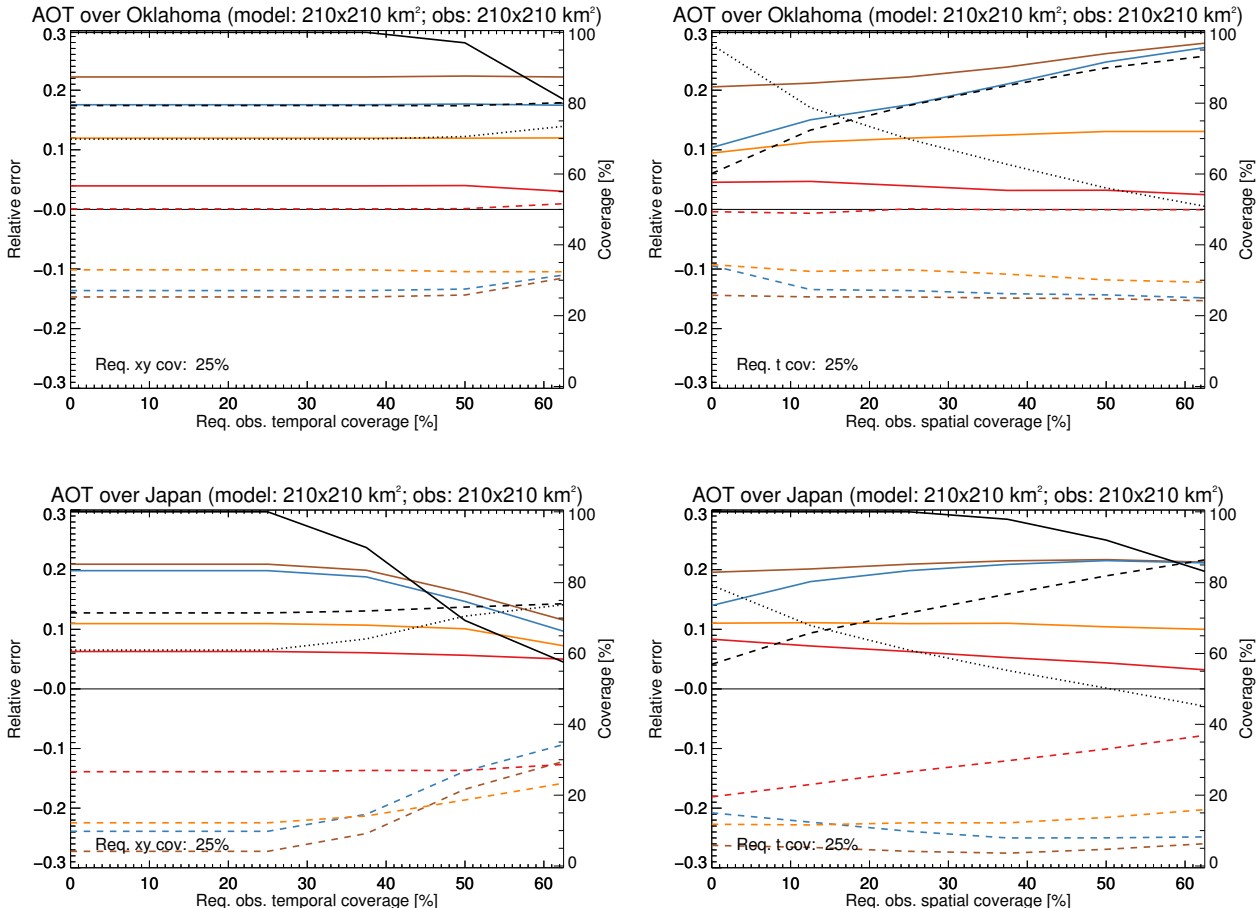

**Figure 13.** Monthly mean (dashed) and RMS (solid) of representation errors for an imager on a polar-orbiting satellite as a function of required spatial or temporal coverage of the observations. Results are shown for no collocation (brown), no collocation but using super-observations (dark blue), collocation to the day (orange), and finally model data collocated to the hour (red). Results for a repeat cycle of 1 day. Further explanation in Sect. 3.2.





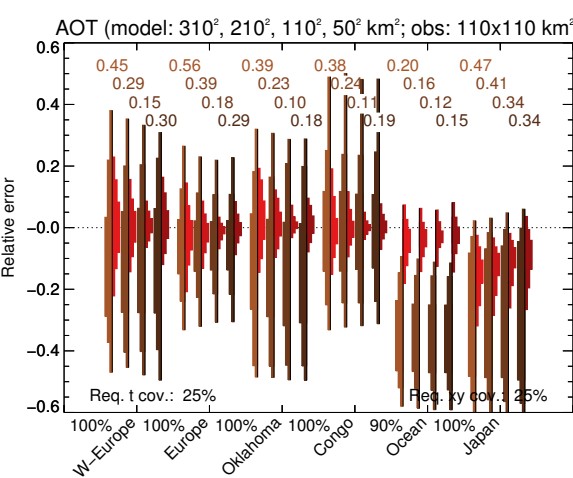

**Figure 14.** Monthly representation errors for an imager on a polar-orbiting satellite at different area sizes but the observations aggregated over $110 \times 110$ km$^2$: no collocation (different shades of brown) and model data collocated to the hour (different shades of red). Results for a repeat cycle of 1 day. Further explanation in Sect. 3.2.





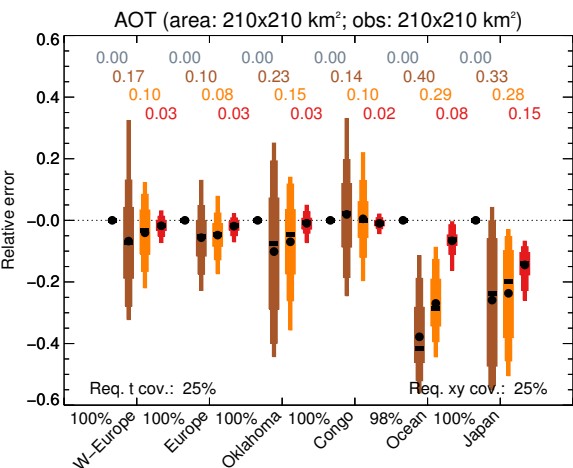

**Figure 15.** Monthly representation errors for an imager on a geostationary satellite after collocation: purely spatial sampling (grey), no collocation (brown), area data collocated to the day of observations (bright orange), and area data collocated to the hour (red). Further explanation in Sect. 3.2.

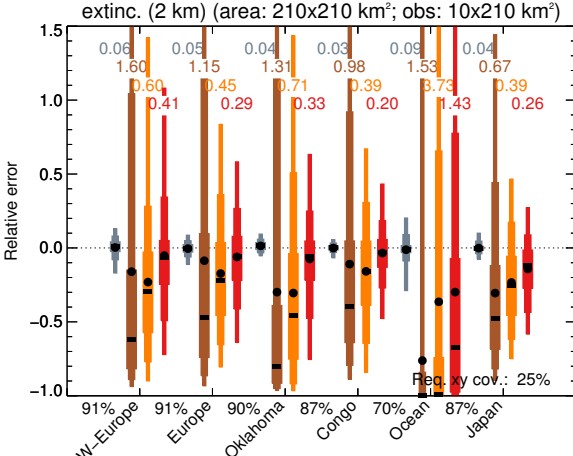

**Figure 16.** Monthly representation errors for a LIDAR on a polar-orbiting satellite after collocation: purely spatial sampling (grey), no collocation (brown), area data collocated to the day of observations (bright orange), and area data collocated to the hour (red). Further explanation in Sect. 3.2.





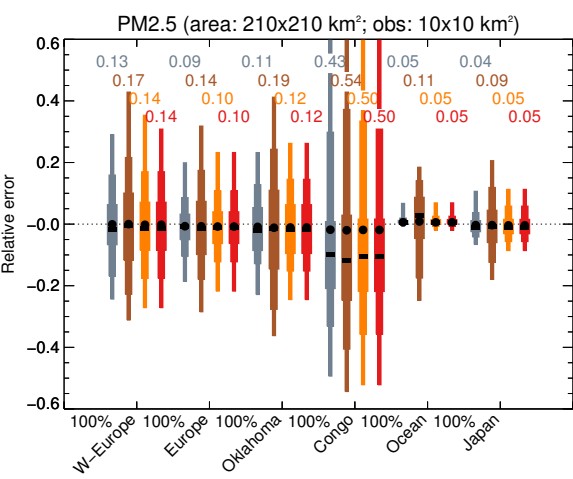

**Figure 17.** Monthly representation errors for an in-situ ground-site after collocation: purely spatial sampling (grey), no collocation (brown), area data collocated to the day of observations (bright orange), and area data collocated to the hour (red). Further explanation in Sect. 3.2.





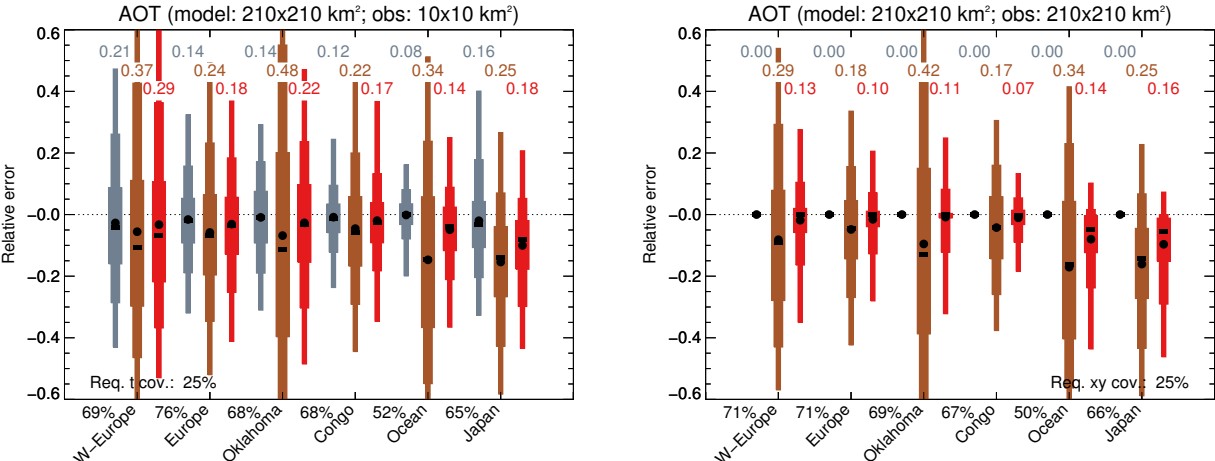

**Figure 18.** Daily representation errors after collocation: purely spatial sampling (grey), no collocation (brown), and model data collocated to the hour (red). The left panel is for a ground-site, the right panel for a satellite with a 1 day repeat cycle. Further explanation in Sect. 3.2.

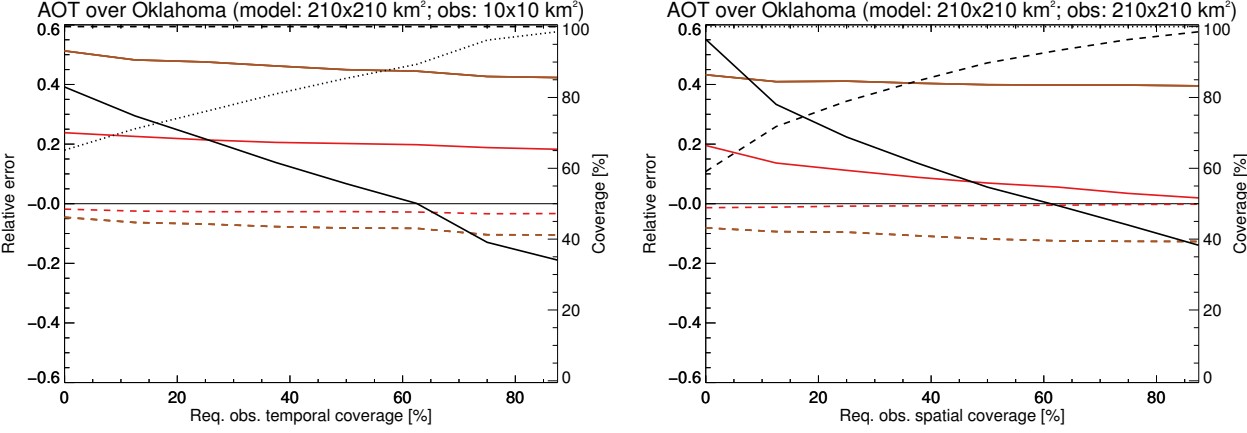

**Figure 19.** Daily representation errors for remote sensing instruments as a function of required coverage. Results shown for no collocation (brown), and area data collocated to the hour (red). The left panel is for a ground-site, the right panel for an imager on a polar-orbiting satellite with a 1 day repeat cycle. Further explanation in Sect. 3.2.





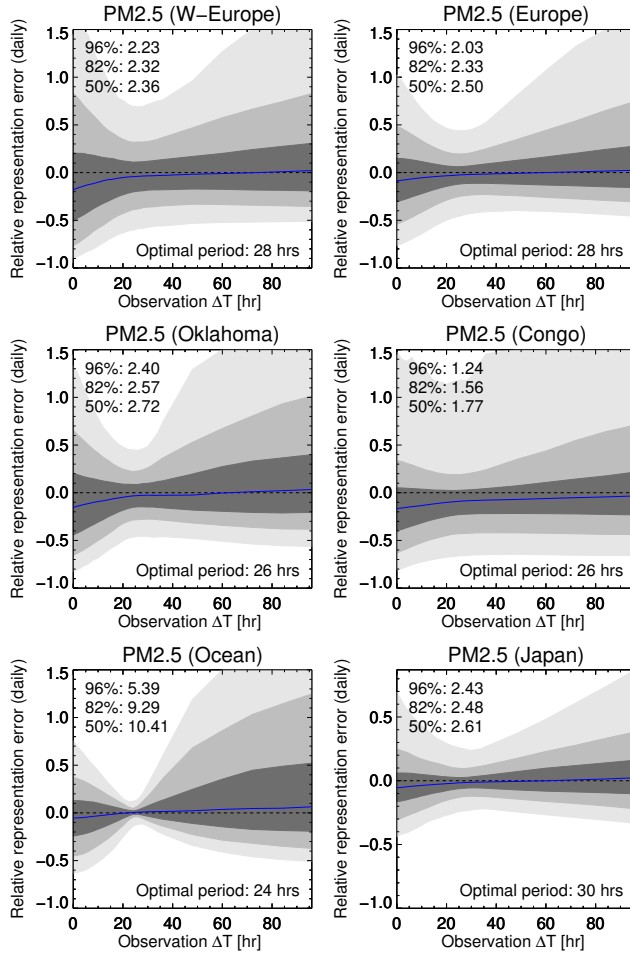

**Figure 20.** Daily representation errors as a function of averaging period $\Delta T$ used for surface PM25 observations. In the top-left corner, the ratio of $q_{98} - q_2, q_{91} - q_9$ and $q_{75} - q_{25}$ for $\Delta T = 0$ to optimal $\Delta T$ is given. Results for a $210 \times 210$ km grid-box. Further explanation in Sec. 3.2.




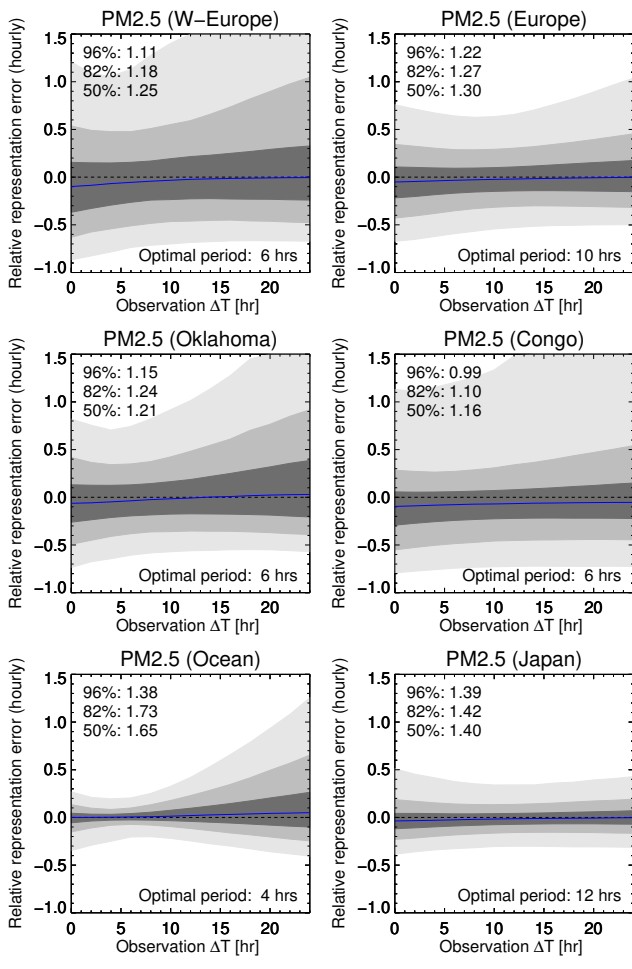

**Figure 21.** Hourly representation errors as a function of averaging period $\Delta T$ used for surface PM25 observations. In the top-left corner, the ratio of $q_{98} - q_2, q_{91} - q_9$ and $q_{75} - q_{25}$ for $\Delta T = 0$ to optimal $\Delta T$ is given. Results for a $210 \times 210$ km grid-box. Further explanation in Sec. 3.2

.





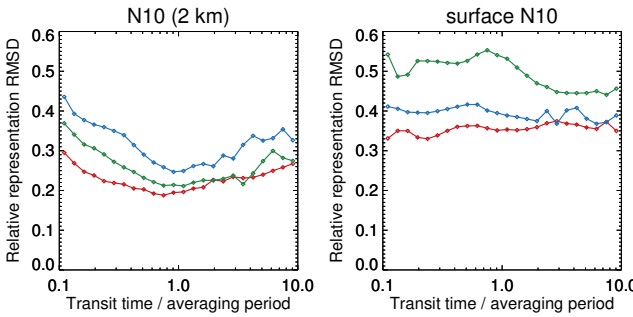

**Figure 22.** Relative representation RSMD for N10 measurements as a function of transit time over averaging period, for W-Europe (red), Oklahoma (blue) and Congo (green). Further explanation in Sec. 3.2.

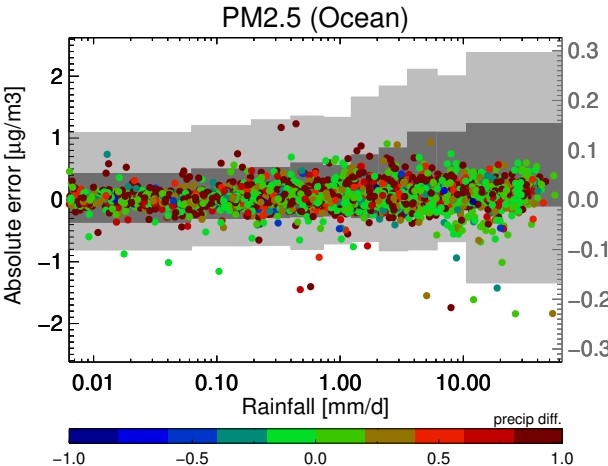

**Figure 23.** Impact on daily representation errors from precipitation. The symbols use the left-hand axis (colours indicate relative difference in precipitation between observation and wider area), the grey quantile boxes the right-hand axis. Results for a $210 \times 210$ km grid-box for Ocean.





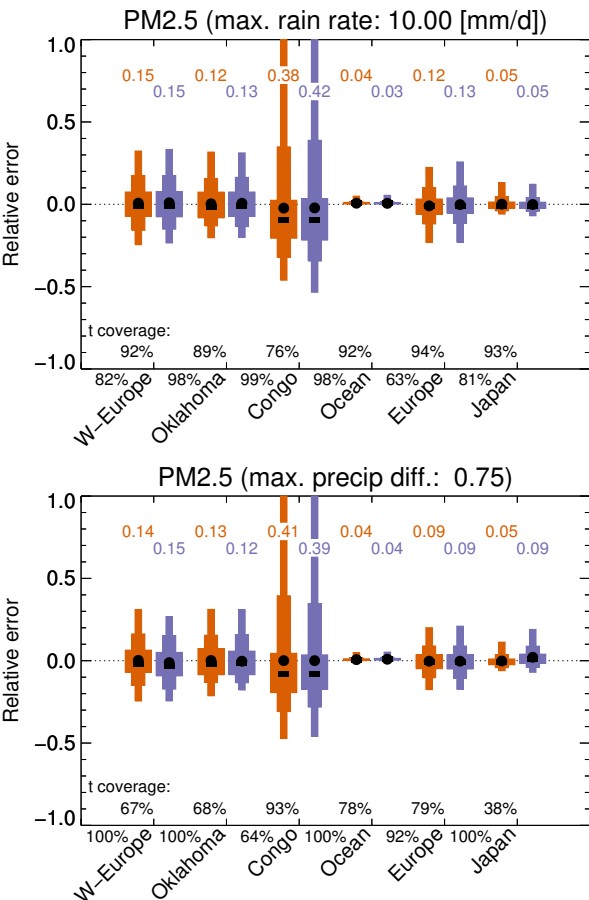

**Figure 24.** Impact on monthly representation errors from filtering out precipitation events. Orange box-whiskers show errors when all data is used, purple box-whiskers show errors when precipitation events are discard (top: daily precipitation > 10 mm/d; bottom: daily precipiation difference > 0.75). Only locations where this maximum was exceeded at least once were used in the statistics. Results for a $210 \times 210$ km grid-box. Further explanation in Sect. 3.2.



**Table 1.** Simulations analysed in this study

| region | size [km$^2$] | period | model | scheme | comments |
|---|---|---|---|---|---|
| W-Europe | $1280 \times 1280$ | May 2008 | WRF-Chem | MADE | 2-moments modal |
| Oklahoma | $1190 \times 1190$ | March 2007 | WRF-Chem | MADE | 2-moments modal |
| Congo | $2090 \times 2090$ | March 2007 | WRF-Chem | MADE | 2-moments modal |
| Ocean | $1270 \times 1270$ | March 2007 | WRF-Chem | GOCART | mass bulk |
| Europe | $4000 \times 3100$ | January - June 2008 | EMEP | | mass bulk |
| Japan | $1500 \times 1250$ | August 2007 | NICAM | SPRINTARS | mass bulk |

**Table 2.** Simulated observables

| | AOT | AE | SSA | extinction | PM$_{2.5}$ | BC conc. | N10, N50 | CCN |
|---|---|---|---|---|---|---|---|---|
| WRF-Chem MADE | ✓ | ✓ | ✓ | ✓ | ✓ | ✓ | ✓ | ✓ |
| WRF-Chem GOCART | ✓ | ✓ | ✓ | ✓ | ✓ | | | |
| EMEP | ✓ | ✓ | ✓ | ✓ | ✓ | ✓ | | |
| NICAM-SPRINTARS | ✓ | ✓ | ✓ | ✓ | ✓ | ✓ | | |





**Table 3.** Semi-annual relative representation errors for ACTRIS sites

|  | Harwell | Hohenpeissenberg | Hyytiala | MtCimone | Preila | Puy de Dome |
|---|---|---|---|---|---|---|
| longitude | -1.32 | 11.01 | 24.29 | 10.68 | 21.04 | 2.97 |
| latitude | 51.57 | 47.80 | 61.85 | 44.17 | 55.21 | 45.77 |
| altitude [m] | 60 | 985 | 181 | 2165 | 6 | 1465 |
|  |  |  |  |  |  |  |
| daily surf. BC [%] | 23.2 | 20.1 | 13.1 | 54.1 | 24.2 | 52.1 |
| Jan-Jun surf. BC [%] | -1.4 | -9.9 | -0.3 | -53.7 | -4.5 | 30.8 |
|  |  |  |  |  |  |  |
| daily AOT [%] | 23.2 | 27.7 | 28.7 | 38.0 | 29.1 | 27.6 |
| 6-month AOT [%] | -27.9 | -23.7 | -38.0 | -29.9 | -34.8 | -11.2 |
|  |  |  |  |  |  |  |
| With collocation |  |  |  |  |  |  |
| daily AOT [%] | 12.2 | 21.2 | 12.8 | 33.1 | 17.0 | 18.3 |
| 6-month AOT [%] | -1.7 | -8.9 | -1.9 | -25.4 | -3.5 | -6.2 |



**Table 4.** Optimal averaging periods for ground-site measurements used to represent a $210 \times 210$ km$^2$ area (daily). The colours indicate an increase of representation RMSD representation by less than 5%, less than 10% or less than 20% when using the recommend period of 24 hours instead.

|  | W-Europe | Oklahoma | Congo | Ocean | Europe | Japan |
|---|---|---|---|---|---|---|
| AOT | 30 | 26 | 26 | 24 | 28 | 28 |
| AE | 32 | 26 | 28 | 24 | 28 | 30 |
| SSA | 32 | 26 | 28 |  | 30 | 30 |
| PM$_{2.5}$ | 28 | 26 | 26 | 24 | 28 | 30 |
| surface extinction | 28 | 26 | 26 | 24 | 28 | 30 |
| extinction (h=2km) | 30 | 26 | 26 | 24 | 28 | 28 |
| surface BC conc | 30 | 26 | 26 |  | 28 | 32 |
| BC conc (h=2km) | 28 | 26 | 26 |  |  | 30 |
| surface N10 | 48 | 26 | 24 |  |  |  |
| N10 (h=2km) | 30 | 26 | 26 |  |  |  |
| surface N50 | 34 | 26 | 26 |  |  |  |
| N50 (h=2km) | 28 | 26 | 26 |  |  |  |

**Table 5.** Optimal averaging periods for ground-site measurements used to represent a $210 \times 210$ km$^2$ area (hourly). The colours indicate an increase of representation RMSD representation by less than 5%, less than 10% or less than 20% when using the recommend period of 6 hours instead.

|  | W-Europe | Oklahoma | Congo | Ocean | Europe | Japan |
|---|---|---|---|---|---|---|
| AOT | 10 | 6 | 8 | 6 | 10 | 10 |
| AE | 10 | 6 | 8 | 6 | 10 | 14 |
| SSA | 10 | 8 | 8 |  | 8 | 14 |
| PM$_{2.5}$ | 6 | 6 | 6 | 4 | 10 | 12 |
| surface extinction | 4 | 4 | 6 | 4 | 8 | 10 |
| extinction (h=2km) | 8 | 4 | 6 | 4 | 8 | 10 |
| surface BC conc | 10 | 4 | 6 |  | 10 | 14 |
| BC conc (h=2km) | 6 | 8 | 8 |  |  | 12 |
| surface N10 | 8 | 2 | 2 |  |  |  |
| N10 (h=2km) | 10 | 6 | 6 |  |  |  |
| surface N50 | 8 | 6 | 6 |  |  |  |
| N50 (h=2km) | 8 | 4 | 8 |  |  |  |