# Peer review of "On the spatio-temporal representativeness of observations"

_Atmospheric Chemistry and Physics, 2017_

## Referee Comment (RC1) · Anonymous Referee #1 · 18 Apr 2017

**Referee report for "*On the spatio-temporal representativeness of observations*",
submitted by Nick Schutgens et al. to ACP.**

Dear authors,

It was with great interest and pleasure that I read your new contribution on the representativeness of observations. As measurement accuracies improve and multi-instrument studies become more and more prevalent, the representativeness of individual and averaged data becomes of increasing importance and this paper therefore addresses a real and acute need. While the general approach (using high-resolution model fields to simulate representativeness errors) is not novel, this paper reports on the first application to estimate the combined spatio-temporal representativeness of aerosol(-related) data and as such clearly deserves publication. Moreover, the manuscript reads very well with a clear structure, few typing errors, and easy-to-read figures. My only truly general remark would be that the work got chopped up into too many (3) papers, leading to some repetition but also requiring the reader to have at hand the other papers, and actually making parts of the previous papers, less than a year old, somewhat obsolete. Of course, this can no longer be changed.

Besides these general thoughts, I have a few **specific concerns and suggestions** for improvements:
1) The title needs to be more specific, clarifying that this paper is about aerosols. The scope of the results presented here does not warrant the current title. For instance, the representativeness of trace gas and other meteorological observations is impacted by many factors not addressed in the current paper (e.g. the photochemical diurnal cycle of NOx, Ozone depletion in the polar vortices, small-scale variability in the water vapour field, the diurnal variability of temperature…). See also the next comment.
2) Even though this paper is about aerosols, the introduction could/should touch more broadly upon the literature that exists in other atmospheric domains, also outside the assimilation context. For instance: the seminal workshop report by Nappo et al. (1982, Workshop on the representativeness of meteorological observations, held in Boulder in June 1981), papers on the sampling uncertainty and representativeness errors in Ozone observations (Sofieva et al., AMT, 2014b; Section 3.1 in Coldewey-Egbers, AMT, 2015) etc.
3) Even though some references are provided in the introduction to empirical estimates of aerosol spatio-temporal variability and some caveats are given in the conclusions, it would be good to have a paragraph providing some quantitative information on the known/expected variability within a model pixel, i.e. variability at scales smaller than 10km and 1hour. This is in particular relevant to assess the completeness of the error estimates for in situ measurements.

Below are some more highly specific or **technical comments**.
1) Section 2.1, 1$^{st}$ sentence seems redundant (basically saying that the simulated fields are those that were simulated).
2) Section 2, more general: are the hourly data hourly averages or hourly snapshots?

3) Page 5, line 10 (about the observational sampling): in reality, the observations don't occur exactly on the x,y,t of the model. Does that matter, and if not, why not?

4) Page 5, line 14: temporal collocation can of course also be used when comparing different measurement (e.g. in situ versus satellite, so not only in observation-model comparisons), so the scope of these results is wider than is portrayed in the paper.

5) Section 3, more general: why only look at temporal collocation and not spatial collocation? Clouds could also be dealt with using spatial masks instead of temporal collocation. For orography, a spatial mask would be the only solution.

6) Page 7, section 3.4, 1st sentence: Fig. 6 is the first box-whisker plot, not Fig.7

7) Page 8, section 4, 1st sentence: Maybe add "only" to the beginning of the sentence: "Only the EMEP…"

8) Section 4 (and subsequent, more general): why this particular choice of 210x210km2? Most current gridded data sets, whether from satellite or model, have better resolution than that.

9) Page 8, line 9: explain why day-light AOT is lower than average AOT, if known.

10) Page 8, line 21: how come? Please explain briefly.

11) Section 5: again, why 210x210?

12) Page 8, line 29-30: is it known why cloudy AOT is larger than clear-sky AOT for these regions?

13) Page 9, line 12. Although you make it explicit later in the paper (in section 5.3), I think it would be good to state earlier on that the strong effect of temporal sampling/the huge gains with temporal collocation, are all about clouds.

14) Page 9, line 14: satellites -> satellite

15) Page 9, line 19: you point out the similar errors between a ground-site and a satellite sounder with a repeat cycle of 1 day. That may be true for the average size of the errors, but the spatio-temporal pattern of those errors should be vastly different, no? The paper contains lots of box-whisker plots summarizing the statistical properties of the representation errors. It would perhaps be nice to see some more maps (like Figs 3 and 4) to be able to judge the spatial patterns of the representation errors. This is to be seen as just a suggestion: if the authors don't see value in that, they can perhaps just include a statement to explain why no further maps are shown.

16) Page 9, line 32: due to -> obtained after temporal

17) Page 12, line 16: please explain somewhere what N10 is.

18) Page 14, line 4-5 (Section 9.2). You state: "The number of observations used in constructing monthly averages cannot be used to control representation errors". I don't understand where this conclusion comes from (which probably indicates I misunderstood something earlier on). I can hardly believe this to be correct: surely a monthly average based on a measurement every day of the month will lead to a better estimate of the monthly mean than an average based on just 1 measurement?

Looking forward to reading the final paper,
A referee

---

## Referee Comment (RC2) · Anonymous Referee #2 · 19 Apr 2017

**Referee report: „On the spatio-temporal representativeness of observations", by Nick Schutgens et al.**

**General comments**

In this study the authors analyze the spatio-temporal representation errors of aerosol data for a variety of observation methods (incl. ground-based and remote sensing) using high resolution model data. This analysis is important and relevant because long-term climate studies and model evaluation rely on the use of L3 data in which the construction of the L3 data may be significantly impacted by the discontinuous spatio-temporal sampling of the observations. However, some of the findings have been presented already in the previous papers S16a and S16b. The paper is written in a clear and concise manner. I recommend publication after minor revisions.

**Minor / Specific comments**

Introduction: I would suggest to add a reference related to representation errors in ozone observations, e.g., Sofieva, V. F., Kalakoski, N., Päivärinta, S.-M., Tamminen, J., Laine, M., and Froidevaux, L.: On sampling uncertainty of satellite ozone profile measurements, Atmos. Meas. Tech., 7, 1891-1900, doi:10.5194/amt-7-1891-2014, 2014.

Page 3, Section 2.1: Please explain N10/N50 and introduce "BC" as abbreviation for black carbon (used later on in the paper).

Sections 3.2 - 3.5: I would suggest to merge the description of the different figures into one subsection.

Page 8, lines 9/10: Why is clear sky day-light AOT lower than average AOT?

Page 11, Sec. 6.2: Are the numbers the errors due to "purely spatial sampling"?

Page 14, Sec. 9.3: Please add a comment here that you find similar results for polar orbiting satellites and geostationary satellites. At least for me this was a bit surprising as I expected lower errors for the geostationary satellite observations due to multiple views per day (instead of one measurement per day for the LEO).

Page 15, line 30: Not sure whether I can follow conclusion 3). Could you please add an explanation here. Like referee #1 (her/his comment no. 18) I think that estimates of the monthly mean will improve with increasing number of observations.

Page 16, lines 7/8: You say that the results were robust across the regions, but what about the selected months? Did you analyze the natural variability of the observables as a function of month? Do you think that the selected months are representative for the whole year / other years? Errors may increase/decrease significantly if natural variability is different for different months.

Fig. 2: I find it difficult to identify the blue line. Is it possible to show only one red line (e.g., mean/median + std.dev. of all observations 2000-2010)?

**Technical corrections**

Page 22, Fig. 5, caption: "210 x 210 km" -> "210 x 210 km$^2$"

Page 25, Fig. 9, title: "obs: 210 x 10 km$^2$" -> "obs: 10 x 10 km$^2$"

Page 31, Fig. 16, title: "obs: 10 x 210 km$^2$" -> "obs: 210 x 210 km$^2$"

Page 34, Fig. 20, caption: "PM25" -> "PM2.5" and "km" -> "km$^2$"

Page 35, Fig. 21, caption: "PM25" -> "PM2.5" and "km" -> "km$^2$"

Page 36, Fig. 23, caption: "km" -> "km$^2$"

Page 37, Fig. 24, caption: "km" -> "km$^2$"

---

## Author Comment (AC1) · 5 May 2017

**Reply to reviewer 1**

We would like to thank the reviewer for their time, especially since this is a long paper, and useful & considerate comments. Their comments are repeated below in italic, followed by our answers.

*My only truly general remark would be that the work got chopped up into too many (3) papers, leading to some repetition but also requiring the reader to have at hand the other papers, and actually making parts of the previous papers, less than a year old, somewhat obsolete.*

These papers bring out different aspects of sampling issues, for a variety of observing systems and observables. While there is a thematic overlap, we feel the overlap is small content-wise:
- S16a concerns spatial sampling, in a model evaluation context. It uses regional model data and assumes highly localised observations. It really is a study of sampling errors for continuously measuring (in-situ) ground-sites or incidental flight campaigns. It shows that different observations can lead to very different sampling errors. It includes a lot of sensitivity studies for different strategies in comparing a global model to the observations.
- S16b concerns temporal sampling, in a model evaluation context. It used global model data and real remotely sensed observations. It compared sampling errors to actual model errors and showed them to be of similar magnitude. It also showed that models compared better with real observations after temporal collocation. It showed that the sampling error for visual remote sensing data would depend on longitude when using daily model data.
- The current study concerns spatio-temporal sampling in a general sense. It uses regional model data and a separate model for (idealised) observational spatio-temporal sampling. It allows the study of sampling issues in satellite L3 data (not possible with S16a or S16b) and provides realistic estimates for representation errors after temporal collocation (again not possible with S16a or S16b).

The paragraph describing S16a and S16b (p 2 line 20-28, in the introduction) has been rewritten to clarify this.

**Specific concerns/suggestions**

*The title needs to be more specific clarifying that this paper is about aerosols. The scope of the results presented here does not warrant the current title.*

The reviewer is correct in suggesting that the magnitude of representation errors may be very different for observations that we have not considered. Our literature study suggests that compared to aerosol observations, representation errors in e.g. ozone, solar surface radiation or water vapour column are relatively small (we are not saying they are insignificant!). Although we only consider aerosol measurements, it should be noted these are very diverse in nature and often the result of very different processes (see also S16a). Consequently, we believe that our paper holds interest for other fields: 1) it provides a paradigm for studying these errors (we have not encountered the combined issue of spatio-temporal sampling in the literature before); 2) it shows how representation errors depend on sampling strategies and averaging

procedures. We find it hard to believe that this will be fundamentally different for other observables.

The nature of ACP and the content of the abstract make the limitations in our paper quite clear, but a title should also be used to advertise a particular topic. We suggest to keep the title as it is.

*Even though this paper is about aerosols the introduction could/should touch more broadly upon the literature that exists in other atmospheric domains also outside the assimilation context.*

We were not aware of the work on representation in ozone measurements. The Nappo report can no longer be found in the BAMS archive (presumably this is a summary only), and our university library staff could not obtain a copy of the full report. While we have not been able to obtain the Nappo et al. report, the other papers mentioned by the reviewer will be referenced.

We also suggest Lin et al. 2015 GRL "Revisiting the evidence of increasing springtime ozone mixing ratios in the free troposphere over Western North America" and Boersma et al. 2016 GMD "Representativeness errors in comparing chemistry transport and chemistry climate models with satellite UV-Vis tropospheric column retrievals" to add to the paragraph describing representation studies in climate variables, surface radiation, SST and water vapour measurements.

*Even though some references are provided in the introduction to empirical estimates of aerosol spatio-temporal variability and some caveats are given in the conclusions, it would be good to have a paragraph providing some quantitative information on the known/expected variability within a model pixel, i.e. variability at scales smaller than 10km and 1hour. This is in particular relevant to assess the completeness of the error estimates for in situ measurements.*

Unfortunately, we don't know of any beyond what we already mention (e.g. Anderson et al 2003). Since most atmospheric variables show a power law distribution when performing a Fourier analysis in space and time (see also Fig 3, S16b), we suggest that variability below 10 km and 1 hour will typically be smaller than that above 10 km and 1 hour. Undoubtedly exceptions will exist.

**Technical comments**

*Section 2.1,1st sentence seems redundant (basically saying that the simulated fields are those that were simulated)*

The sentence reads: "The simulated fields examined in this paper are, for obvious reasons, all observables". It is a slightly trivial sentence but simulated fields are not necessarily observable.

*Section 2, more general: are the hourly data hourly averages or hourly snapshots?*

They are snapshots, except for the precipitation which are accumulated fields. This will be added to Sect 2.

*Page 5,line 10 (about the observational sampling): in reality, the observations don't occur exactly on the x,y,t of the model. Does that matter, and if not, why not?*

This is an unavoidable simplification. If the high-resolution runs were at 100 m instead of 10 km, we would be able to position in-situ observations even more accurately compared to the larger area. However, given the large size of the represented area (210 by 210 km2), we expect an error of at most 10 km in the location of an observation to be negligible in impact. See also our answer to the reviewer's third specific suggestion and the expected variation at 10km scales.

*Page 5, line 14: temporal collocation can of course also be used when comparing different measurement (e.g. in situ versus satellite, so not only in observation-model comparisons), so the scope of these results is wider than is portrayed in the paper.*

Indeed. This is why we tried to avoid mention of model evaluation in the current paper. (Note that the previous paper was titled: "Will a perfect model agree with perfect observations? The impact of spatial sampling"). The representation issue is also (or even doubly) important when comparing different observational datasets.

*Section3, more general: why only look at temporal collocation and not spatial collocation? Clouds could also be dealt with using spatial masks instead of temporal collocation. For orography, a spatial mask would be the only solution.*

Possibly we misunderstand the question but if that were possible, wouldn't representation errors (after collocation) be zero by definition? We have assumed that the represented area has a fixed size & shape, either because it represents a model gridbox or because of operational considerations (it is possible to identify regions where the field values strongly correlate with the observations, e.g. Piersanti et al. APR 2015. But those regions will vary from day to day and location to location, making this approach unpractical). Note that even the influence of orography is not clear cut, as usually wind-flows combine with orography to cause the representation issues.

*Page 7, section 3.4, 1ˢᵗ sentence: Fig. 6 is the first box-whisker plot, not Fig.7*

Corrected.

*Page 8, section 4, 1st sentence: Maybe add "only" to the beginning of the sentence: "Only the EMEP..."*

Agreed.

*Section 4 (and subsequent, more general): why this particular choice of 210x210km2? Most current gridded data sets, whether from satellite or model, have better resolution than that.*

It shows our interest in model evaluation (most state-of-the-art global aerosol models still run at fairly low resolutions). A typical T63 grid translates into a 210

by 210 km2 box at the equator. Note that we have included analysis of representation errors for smaller areas (and see also S16b for more detail on this), in particular 110 by 110 km2 (1 by 1 degree at the equator).

*Page 8, line 9: explain why day-light AOT is lower than average AOT, if known.*

Average day-light AOT is only slightly smaller than night-time AOT (few %) for unknown reasons. However, average clear-sky AOT is decidedly smaller than cloudy AOT (mostly due to increased humidity in the cloudy column). Day-light is mentioned in because it is one of two conditions for valid observations. We have replaced 'clear-sky day-light AOT' with 'observable AOT' and added an explanation.

*Page 8, line 21: how come? Please explain briefly.*

We assume the reviewer would like to know why EMEP shows smaller representation errors than WRF-Chem. We discuss this in S16a in some detail. Briefly, it is impossible to say why without a separate study into why EMEP and WRF-Chem differ in the first place. We noted that magnitudes and spatial patterns agreed nicely, giving us confidence in the use of these models.

*Section 5: again, why 210x210?*

See before.

*Page 8, line 29-30: is it known why cloudy AOT is larger than clear-sky AOT for these regions?*

Please see explanation before (the question regarding page 8, line 9).

*Page 9, line 12. Although you make it explicit later in the paper (in section 5.3), I think it would be good to state earlier on that the strong effect of temporal sampling/the huge gains with temporal collocation, are all about clouds.*

This is true for ground-sites, polar orbiting satellites with short repeat cycles or geo-stationary satellites. But for polar-orbiting satellites with long repeat cycles (e.g. LIDAR), the operational cycle (revisit time) is far more important.

*Page 9, line 14: satellites -> satellite*

Corrected.

*Page 9, line 19: you point out the similar errors between a ground-site and a satellite sounder with a repeat cycle of 1 day. That may be true for the average size of the errors, but the spatio-temporal pattern of those errors should be vastly different, no? The paper contains lots of box-whisker plots summarizing the statistical properties of the representation errors. It would perhaps be nice to see some more maps (like Figs 3 and 4) to be able to judge the spatial patterns of the representation errors. This is to be seen as just a suggestion: if the authors don't see*

*value in that, they can perhaps just include a statement to explain why no further maps are shown.*

The spatial pattern of those representation errors is somewhat different but not too much. One of our conclusions is that monthly data like L3 suffers mostly from temporal sampling issues (no observations at night time or cloudy skies). This will be fairly similar for a ground-site and a satellite. We show results for Oklahoma below:

[Figure]

We suggest to add one of these figures to the paper, because they show that although overall statistics (box-whisker plots) seem unbiased, strong bias may exist in separate parts of the region.

*Page 9, line 32: due to -> obtained after temporal*

Agreed.

*Page 12, line 16: please explain somewhere what N10 is.*

This is now explained in Sect. 2.1: "N10 and N50, number densities for particles with diameters exceeding 10 resp. 50 nm"

*Page 14, line 4-5 (Section 9.2). You state: "The number of observations used in constructing monthly averages cannot be used to control representation errors". I don't understand where this conclusion comes from (which probably indicates I misunderstood something earlier on). I can hardly believe this to be correct: surely a monthly average based on a measurement every day of the month will lead to a better estimate of the monthly mean than an average based on just 1 measurement?*

While we agree with the reviewer's point point, we wanted to study

representation errors in the context of realistically achievable number of observations. The reviewer's example is fairly abstract as both cases seldom occur.

The relevant figure is Fig. 8 that shows monthly representation errors for ground-sites as a function of *required* temporal coverage. *Actual* temporal coverage (or the number of observations) will always be higher and is shown by the black dotted line (right axis). The brown line (left axis) represents representation errors when data are *not* collocated (which is what our statement was about). Note that an increased number of observations *may* reduce representation errors, as is shown for Japan. However, for Oklahoma (and most other regions) this error hardly changes with the number of observations. A combination of strong temporal variation throughout the day, and different spatial sampling of the ground-site and represented area prevents an increasing number of observations to reduce representation errors.

Strictly speaking our statement should have read "Using a minimum required number of observations cannot be relied upon to control representation errors." The text will be changed.

---

## Author Comment (AC2) · 5 May 2017

**Reply to reviewer 2**

We would like to thank the reviewer for their time, especially since this is a long paper, and useful & considerate comments. Their comments are repeated below in italic, followed by our answers.

**General comments**

*However, some of the findings have been presented already in the previous papers S16a and S16b.*

These papers bring out different aspects of sampling issues, for a variety of observing systems and observables. While there is a thematic overlap, we feel the overlap is small content-wise:
- S16a concerns spatial sampling, in a model evaluation context. It uses regional model data and assumes highly localised observations. It really is a study of sampling errors for continuously measuring (in-situ) ground-sites or incidental flight campaigns. It shows that different observations can lead to very different sampling errors. It includes a lot of sensitivity studies for different strategies in comparing a global model to the observations.
- S16b concerns temporal sampling, in a model evaluation context. It used global model data and real remotely sensed observations. It compared sampling errors to actual model errors and showed them to be of similar magnitude. It also showed that models compared better with real observations after temporal collocation. It showed that the sampling error for VIS remote sensing data would depend on longitude when using daily model data.
- The current study concerns spatio-temporal sampling in a general sense. It uses regional model data and a separate model for the idealised observational spatio-temporal sampling. It allows the study of sampling issues in satellite L3 data (not possible with S16a or S16b) and provides realistic estimates for representation errors after temporal collocation (again not possible with S16a or S16b).

The paragraph describing S16a and S16b (p 2 line 20-28, in the introduction) has been rewritten to clarify this.

**Minor/specific comments**

*Introduction: I would suggest to add a reference related to representation errors in ozone observations, e.g., Sofieva, V. F., Kalakoski, N., Päivärinta, S.-M., Tamminen, J., Laine, M., and Froidevaux, L.: On sampling uncertainty of satellite ozone profile measurements, Atmos. Meas. Tech., 7, 1891-1900, doi:10.5194/amt-7-1891-2014, 2014.*

This is an interesting paper that the other reviewer suggested as well. We were not familiar with it but have now added it to the introduction.

*Page 3, Section 2.1: Please explain N10/N50 and introduce "BC" as abbreviation for black carbon (used later on in the paper).*

Agreed.

*Sections 3.2 - 3.5: I would suggest to merge the description of the different figures into one subsection.*

We agree the page layout does look a bit awkward, but the benefit (we hope) of the subsections is that readers will be able to quickly look up the description relevant to a particular graph. We suggest to not change this.

*Page 8, lines 9/10: Why is clear sky day-light AOT lower than average AOT?*

Average day-light AOT is only slightly smaller than night-time AOT (few %) for unknown reasons. However, average clear-sky AOT is decidedly smaller than cloudy AOT (mostly due to increased humidity in the cloudy column). Day-light is mentioned in because it is one of two conditions for valid observations. We have replaced 'clear-sky day-light AOT' with 'observable AOT' and added an explanation

*Page 11, Sec. 6.2: Are the numbers the errors due to "purely spatial sampling"?*

Indeed. Our assumption is that such in-situ ground-sites measure continuously, at least for the duration of a day. Consequently, daily representation errors are purely due to spatial sampling.

*Page 14, Sec. 9.3: Please add a comment here that you find similar results for polar orbiting satellites and geostationary satellites. At least for me this was a bit surprising as I expected lower errors for the geostationary satellite observations due to multiple views per day (instead of one measurement per day for the LEO).*

Agreed.

While this result seems counter-intuitive, it is a consequence of 1) temporal variation throughout the day that even the GEO sensor can not observe; 2) cloud masking over 210 by 210 km2 that prevents observation of the entire area by the GEO sensor. These two causes contribute in roughly equal measure to the final representation error for geostationary sensors that can only observe during the day.

Although we do not mention this in the paper, we considered the case of a physically impossible observing system: a geostationary satellite that can observe during both day and night. For areas without (!) clouds, daily representation errors are indeed zero as expected.

*Page 15, line 30: Not sure whether I can follow conclusion 3). Could you please add an explanation here. Like referee #1 (her/his comment no. 18) I think that estimates of the monthly mean will improve with increasing number of observations.*

While we agree with the reviewer's point, we wanted to study representation errors in the context of realistically achievable number of observations. Reviewer's #1 example is fairly abstract as both cases seldom occur.

The relevant figure is Fig. 8 which shows monthly representation errors for ground-sites as a function of *required* temporal coverage. *Actual* temporal coverage (or the number of observations) will always be higher and is shown by the black dotted line (right axis). The brown line (left axis) represents

representation errors when data are *not* collocated (which is what our statement was about). Note that an increased number of observations *may* reduce representation errors, as is shown for Japan. However, for Oklahoma (and most other regions) this error hardly changes with the number of observations. A combination of strong temporal variation throughout the day, and different spatial sampling of the ground-site and represented area prevents an increasing number of observations to reduce representation errors.

Strictly speaking our statement should have read "Using a minimum required number of observations cannot be relied upon to control representation errors." The text will be changed.

*Page 16, lines 7/8: You say that the results were robust across the regions, but what about the selected months? Did you analyze the natural variability of the observables as a function of month? Do you think that the selected months are representative for the whole year / other years? Errors may increase/decrease significantly if natural variability is different for different months.*

Errors will increase or decrease with variability but we never saw a significant change (for argument's sake here defined as a changed by a factor 3, ie. a 30% error becoming a 10% or 90% error). Also, "robust" referred to the second part of the previous sentence ("their behaviour (e.g. impact from sampling or collocation)"). We accept there will be changes in exact error values. We have rephrased to improve clarity.

*Fig. 2: I find it difficult to identify the blue line. Is it possible to show only one red line (e.g., mean/median + std.dev. of all observations 2000-2010)?*

This figure will be recreated, with the blue line more prominent.

**Technical corrections**

*Page 22, Fig. 5, caption: "210 x 210 km" -> "210 x 210 km$_2$"*

*Page 25, Fig. 9, title: "obs: 210 x 10 km$_2$" -> "obs: 10 x 10 km$_2$"*

*Page 31, Fig. 16, title: "obs: 10 x 210 km$_2$" -> "obs: 210 x 210 km$_2$"*

*Page 34, Fig. 20, caption: "PM25" -> "PM2.5" and "km" -> "km$_2$"*

*Page 35, Fig. 21, caption: "PM25" -> "PM2.5" and "km" -> "km$_2$"*

*Page 36, Fig. 23, caption: "km" -> "km$_2$"*

*Page 37, Fig. 24, caption: "km" -> "km$_2$"*

Most of corrections will be implemented in the final paper. Note that in Fig 16, our caption is correct: the LIDAR sweeps out a narrow transect (curtain), represented by a 10 x 210 km2 area.